# ezSingleCell: an integrated one-stop single-cell and spatial omics analysis platform for bench scientists

Raman Sethi [1], Kok Siong Ang [2], Mengwei Li[2], Yahui Long [2], Jingjing Ling[3] & Jinmiao Chen [1,2,4] ✉

ezSingleCell is an interactive and easy-to-use application for analysing various single-cell and spatial omics data types without requiring prior programing knowledge. It combines the best-performing publicly available methods for in-depth data analysis, integration, and interactive data visualization. ezSingle-Cell consists of five modules, each designed to be a comprehensive workflow for one data type or task. In addition, ezSingleCell allows crosstalk between different modules within a unified interface. Acceptable input data can be in a variety of formats while the output consists of publication ready figures and tables. In-depth manuals and video tutorials are available to guide users on the analysis workflows and parameter adjustments to suit their study aims. ezSingleCell's streamlined interface can analyse a standard scRNA-seq dataset of 3000 cells in less than five minutes. ezSingleCell is available in two forms: an installation-free web application (https://immunesinglecell.org/ezsc/) or a software package with a shinyApp interface (https://github.com/JinmiaoChenLab/ezSingleCell2) for offline analysis.

Single-cell RNA sequencing (scRNA-seq) has emerged as a powerful technology to acquire gene expression profiles at the single-cell level[1]. This enables novel insights into the heterogeneity of biological systems[2]. Single-cell analysis has also been extended to other omics such as single cell ATAC-seq and CITE-seq[3] to measure chromatin accessibility and proteins, respectively. In addition, advances in spatial transcriptomics are now enabling researchers to probe tissue samples at single-cell resolution while retaining their spatial context[4–7]. The latter capability is crucial in understanding how different cell types are spatially arranged to give rise to a tissue's emergent properties.

The rapid rise of single-cell technologies has led to enormous amounts of data being generated. Alongside, new tools are being developed to analyze the data generated and produce novel biological insights. In 2021, Zappia and Theis reported that the number of cataloged single-cell tools in the scRNA-tools database exceeded a thousand[8]. Currently, two software platforms remain dominant, namely Seurat[9] and Scanpy[10] which are the de facto standards for single-cell analysis in R and Python, respectively. Both require a minimum level of bioinformatics expertise and coding knowledge, thereby presenting a barrier to data analysis for bench scientists. In contrast, tools with intuitive graphic user interfaces would greatly benefit bench scientists that wish to employ single-cell experiments. There are several recently developed web servers from both the research community and commercial companies (Supplementary Table 1), but the majority are limited to transcriptomics analysis only and cannot handle spatial omics, single-cell multi-omics, and single-cell chromatin accessibility (scATAC-seq) derived data in a unified interface. Moreover, these websites mainly provide basic analysis functionalities such

[1]Bioinformatics Institute (BII), Agency for Science, Technology and Research (A*STAR), 30 Biopolis Street, Matrix, Singapore 138671, Singapore. [2]Institute of Molecular and Cell Biology (IMCB), Agency for Science, Technology and Research (A*STAR), 61 Biopolis Drive, Proteos, Singapore 138673, Singapore. [3]Singapore Immunology Network (SIgN), Agency of Science, Technology and Research (A*STAR), 8A Biomedical Grove, Immunos, Singapore 138648, Singapore. [4]Immunology Translational Research Program, Department of Microbiology and Immunology, Yong Loo Lin School of Medicine, National University of Singapore (NUS), 5 Science Drive 2, Blk MD4, Level 3, Singapore 117545, Singapore. ✉e-mail: jinmiao@gmail.com

as quality control, data clustering, and dimension reduction, while more advanced down-stream analyses such as cell type identification and cell-cell interaction are not included in their pipelines. Only ICARUS[11] and Cellar[12] can handle multiple data modalities and provide some downstream analyses like cell-type identification but not others such as cell-cell communication and cell type deconvolution of spatial datasets. Another webserver, SciAp[13] (https://humancellatlas.usegalaxy.eu/), integrates tools from different workflows, including 20 modules from Scanpy, covering data filtering, normalization, variable gene selection, clustering, dimensionality reduction, and trajectory inference methods, but currently handles scRNA-seq data only. In addition, integrative analysis of different data modalities, such as integrating scRNA-seq with spatial transcriptomics, are not possible with current web servers.

To provide a more comprehensive data analysis platform with a user-friendly interface, we present ezSingleCell, an integrated one-stop single-cell and spatial analysis web server for bench scientists (https://immunesinglecell.org/ezsc/). ezSingleCell accepts data input in multiple formats such as text formats (csv and tsv) or 10x Cell Ranger/Space Ranger/Cell Ranger-ATAC output, and returns publication ready figures and tables. ezSingleCell improves on existing single-cell data analysis web servers including SciAp, ICARUS, and CELLAR in the following aspects. Firstly, ezSingleCell encompasses a much larger scope of single-cell data analysis, namely single-cell multi-omics, single-cell ATAC-seq, and spatial transcriptomics. The tools offered for these analyses include our in-house algorithms, GraphST[14] and CELLiD[15], as well as top performing publicly available tools such as Seurat[9], Harmony[16], scVI[17], CellphoneDB[18], MOFA+[19], and Signac[20] as determined by benchmarking studies. Secondly, ezSingleCell provides many advanced analysis capabilities beyond basic analysis pipelines. Analysis options common across all modules include differential gene expression analysis, gene set enrichment, cell type similarity, and cell-cell communication. In addition, ezSingleCell offers module specific analyses such as Peak2GeneLinkage in the scATAC-seq module and cell type deconvolution for spatial datasets. Thirdly, scATAC-seq datasets can be analysed with ezSingleCell, a functionality that most webserver lacks. Fourthly, ezSingleCell can scale up to large datasets using geometric sketching[21]. Geometric sketching subsamples large scRNA-seq datasets spanning a million or more cells while preserving rare cell states. This technique is useful for accelerating clustering, visualization, and integration analyses of large datasets. It has also been observed that geometric sketching is consistently effective at distinguishing biological cell types via clustering. Finally, ezSingleCell allows crosstalk between different analysis modules. For example, processed and analysed single-cell RNA-seq data with annotation can be used to deconvolute cell types in spatial data or to perform label transfer to perform cell type annotation of scATAC-seq data. Currently, there are no webservers that have this ability to link two omics data types together in a unified interface.

## Results
### Overview of ezSingleCell and its advantages over other tools
ezSingleCell consists of five modules, single-cell RNA-Seq (scRNA-seq), single-cell data integration (scIntegration), spatial transcriptomics (ST), single-cell multiomics (scMultiomics), and single-cell ATAC-seq (scATAC-seq) (Fig. 1). Each module offers multiple tools for complete data analysis workflows, starting from data pre-processing to interactive result visualization. For each analysis step, we selected the top performing methods based on our and other benchmarking studies. Both the published and in-house novel algorithms available in ezSingleCell are listed in Supplementary Table 2. For each analysis step, we provide default parameters that are suited for most analyses, but users can also tune them to achieve optimal results. In the scRNA-seq analysis module, users can perform basic analyses such as clustering and differential gene expression analysis, as well as advanced analyses like

cell type identification using our in-house novel algorithm CELLiD or CellTypist[22,23], gene set enrichment analysis (GSEA), and cell-cell communication. We also incorporated the 'clustree'[24] package to aid users in selecting the optimum number of clusters that is relevant to their biological questions. The scIntegration module offers the top four performing data integration algorithms for single-cell transcriptomics, namely Seurat Integration, Harmony, scVI, and fastMNN. We also included quantitative metrics such as iLISI[16] for the users to assess batch integration performance. The Spatial Transcriptomics (ST) module is equipped with Seurat's spatial transcriptomics analysis functionalities and our in-house GraphST algorithm. These tools enable spatial clustering and cell type deconvolution of spatial data acquired with different technology platforms such as 10x Genomics Visium and sub-cellular technologies like 10x Genomics Xenium. For the scMultiomics module, we evaluated all available methods and selected Seurat WNN, and MOFA+[19] for inclusion. These methods can handle multimodal data including CITE-seq (joint scRNA-seq with protein) and 10x Multiome (joint scRNA-seq with ATAC-seq). Finally, the scATAC-seq module offers Signac's[20] functionalities for end-to-end analysis of single-cell chromatin accessibility data, including peak calling, quantification, quality control, dimension reduction, clustering, integration with single-cell gene expression datasets, DNA motif analysis, and interactive visualization.

Here we extensively compare the features of ezSingleCell with publicly available webservers and commercial services. As shown in Tables 1 and 2, most of the web servers (such as SciAp, ASAP, alona, NASQAR, SCTK, and Asc-Seurat) are limited to transcriptomics analysis only. Some web services do provide additional functionalities such as data integration (SCTK 2.0, ICARUS), sc-multiomics (ICARUS, Cellar), scATAC-seq (shinyArchR.UiO), and spatial analysis (Cellar), but they do not offer a unified interface for comprehensive analysis. Existing web servers also don't allow interplay between different analysis modules such as using single-cell RNA-seq data to deconvolute spatial transcriptomics data or using single-cell RNA-seq data to perform cell type label transfer for scATAC-seq data. ezSingleCell also provides advanced downstream analysis functionalities within its modules. Moreover, ezSingleCell provides support for scATAC-seq dataset analysis which most webservers lack. Lastly, ezSingleCell's interface informs users which functions are optional, which need to be run sequentially, and which can be run in parallel.

ezSingleCell is available in two forms: an installation-free web application (https://immunesinglecell.org/ezsc/) (Supplementary Figs. 1, 2), and a software package with a Shiny app interface (https://github.com/JinmiaoChenLab/ezSingleCell2) that can be run on a computer for offline analysis. ezSingleCell's source code is also available on Zenodo (https://doi.org/10.5281/zenodo.10785313).

### ezSingleCell's scRNA-seq module streamlines the analysis of scRNA-seq data
The scRNA-seq analysis module of ezSingleCell relies on Seurat for basic analysis, and other packages / in-house algorithms for advanced analyses such as cell type annotation, gene set enrichment analysis (GSEA), and cell-cell communication (Fig. 2A). In addition to the commonly used functions, ezSingleCell also offers other functionalities such as cell-cycle scoring and regression for the user to mitigate the effects of cell cycle heterogeneity. Here we illustrate the utility of ezSingleCell for scRNA-seq analysis (Figs. 2, 3). As an example, we used the dataset of 2700 peripheral blood mononuclear cells (PBMCs) from the Seurat guided clustering vignette (Supplementary Table 3; Supplementary Dataset 1). ezSingleCell allows users to perform quality control and filter out low quality cells using parameters such as min.genes and min.cells, and visualize the data using violin plots, feature plots, and ridgeline plots (Supplementary Fig. 3A). Users can then perform data pre-processing using either log-normalization or SCTransform and select the desired number of

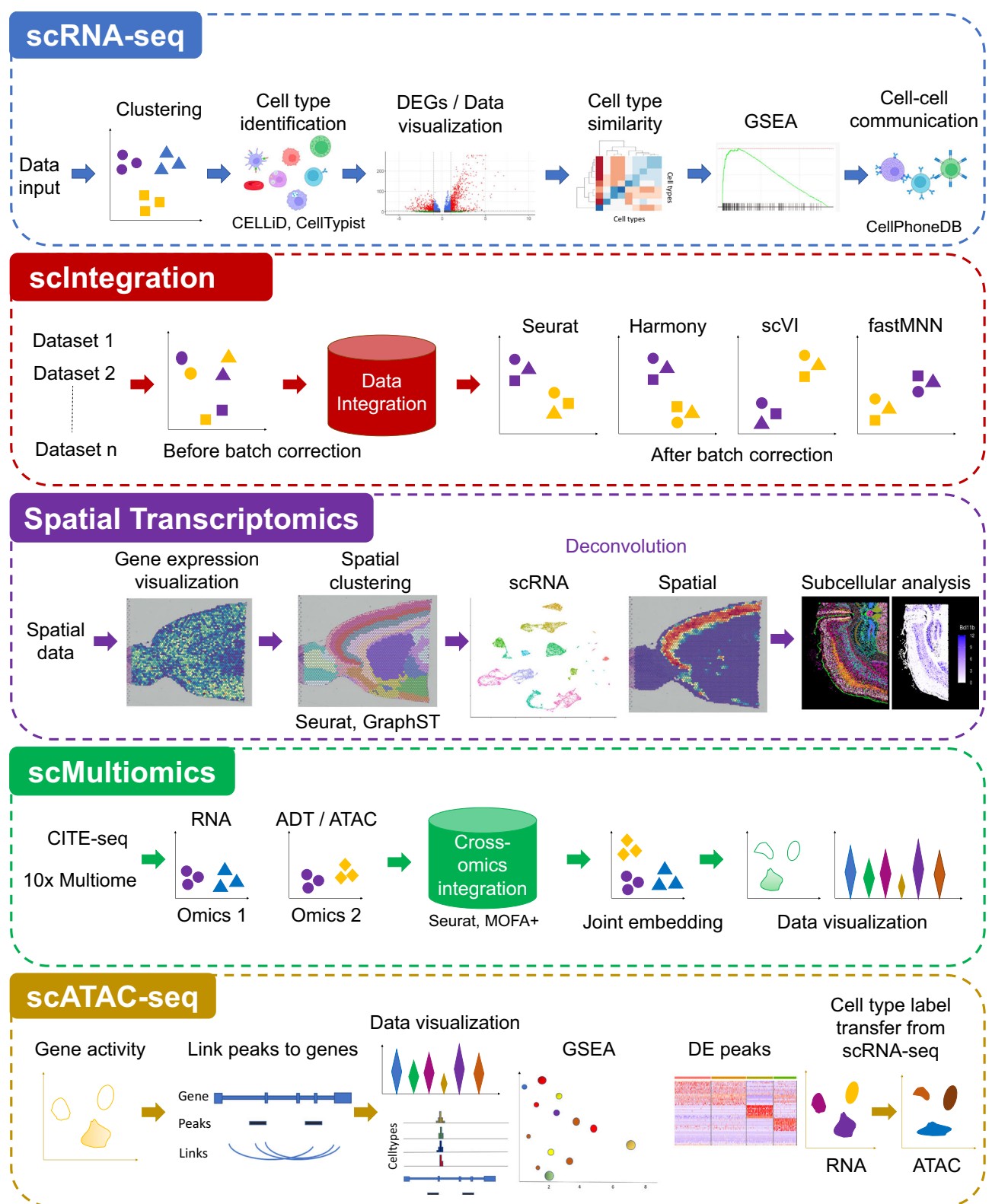

**Fig. 1 | Overview of the ezSingleCell webserver.** ezSingleCell comprises five modules, single-cell RNA-seq (scRNA-seq), single-cell data integration (scIntegration), Spatial Transcriptomics, single-cell Multiomics (scMultiomics), and single-cell ATAC-seq (scATAC-seq). The figure also shows the major tasks that each module can perform along with the tools available in each module. Source data is provided as a Source Data file.

variable features. In this example, we used the default parameter values of min.genes at 200 and min.cells at 3. We then performed data pre-processing with log-normalization, variable feature selection for the top 2000 highly variable genes, and data scaling. This was followed by dimension reduction with Principal Component Analysis (PCA). Cell clustering was then performed using the first 10 PC dimensions, with a *k*-nearest neighbor value of 10 and a clustering resolution of 2 to obtain 15 clusters. The results were visualized using a UMAP plot generated with the PC dimensions 1:10 (Fig. 2B).

**Table 1 | Comparison of ezSingleCell with existing single-cell academic web servers for a variety of tasks in each module (scRNA-seq, scIntegration, scMultiomics, scATAC-seq, and Spatial transcriptomics)**

| Web server | ezSingleCell | ICARUS | ASAP | alona | Cellar | SCiAp | NASQAR | SCTK | Asc-Seurat | ShinyArchR. UiO |
|---|---|---|---|---|---|---|---|---|---|---|
| **scRNA-seq module** | | | | | | | | | | |
| Clustering and dimension reduction | ✓ | ✓ | ✓ | ✓ | ✓ | ✓ | ✓ | ✓ | ✓ | ✗ |
| Cell type identification | ✓ | ✓ | ✓ | ✗ | ✓ | ✓ | ✗ | ✓ | ✗ | ✗ |
| GO/pathway analysis | ✓ | ✓ | ✓ | ✗ | ✓ | ✓ | ✓ | ✓ | ✓ | ✗ |
| Cell-cell communication | ✓ | ✗ | ✗ | ✗ | ✗ | ✗ | ✗ | ✗ | ✗ | ✗ |
| **scIntegration module** | | | | | | | | | | |
| Seurat | ✓ | ✓ | ✗ | ✗ | ✗ | ✗ | ✗ | ✓ | ✗ | ✗ |
| Harmony | ✓ | ✓ | ✗ | ✗ | ✗ | ✗ | ✗ | ✓ | ✗ | ✗ |
| scVI | ✓ | ✗ | ✗ | ✗ | ✗ | ✗ | ✗ | ✗ | ✗ | ✗ |
| fastMNN | ✓ | ✗ | ✗ | ✗ | ✗ | ✗ | ✗ | ✓ | ✗ | ✗ |
| **scMultiomics module** | | | | | | | | | | |
| CITE-Seq | ✓ | ✓ | ✗ | ✗ | ✗ | ✗ | ✗ | ✗ | ✗ | ✗ |
| 10x Multiome | ✓ | ✓ | ✗ | ✗ | ✗ | ✗ | ✗ | ✗ | ✗ | ✗ |
| **scATAC-seq module** | | | | | | | | | | |
| Single-cell ATAC-seq analysis | ✓ | ✗ | ✗ | ✗ | ✗ | ✗ | ✗ | ✗ | ✗ | ✓ |
| **Spatial transcriptomics module** | | | | | | | | | | |
| Clustering | ✓ | ✗ | ✗ | ✗ | ✓ | ✗ | ✗ | ✗ | ✗ | ✗ |
| Deconvolution | ✓ | ✗ | ✗ | ✗ | ✗ | ✗ | ✗ | ✗ | ✗ | ✗ |
| Xenium Analysis | ✓ | ✗ | ✗ | ✗ | ✗ | ✗ | ✗ | ✗ | ✗ | ✗ |
| Free | ✓ | ✓ | ✓ | ✓ | ✓ | ✓ | ✓ | ✓ | ✓ | ✓ |

'✓' and '✗' indicate the presence or absence of a functionality in the web server, respectively.

**Table 2 | Comparison of ezSingleCell with popular commercial services such as 10x Loupe Browser, Partek, and Bioturing for a variety of tasks in each module (scRNA-seq, scIntegration, scMultiomics, scATAC-seq, and Spatial transcriptomics)**

| Web server | ezSingleCell | 10x Loupe Browser | Partek | Bioturing |
|---|---|---|---|---|
| **scRNA-seq module** | | | | |
| Clustering and dimension reduction | ✓ | ✓ | ✓ | ✓ |
| Cell type identification | ✓ | ✗ | ✓ | ✓ |
| GO/pathway analysis | ✓ | ✗ | ✓ | ✓ |
| Cell-cell communication | ✓ | ✗ | ✗ | ✗ |
| **scIntegration module** | | | | |
| Seurat | ✓ | ✗ | ✗ | ✗ |
| Harmony | ✓ | ✗ | ✗ | ✗ |
| scVI | ✓ | ✗ | ✗ | ✗ |
| fastMNN | ✓ | ✗ | ✗ | ✗ |
| **scMultiomics module** | | | | |
| CITE-Seq | ✓ | ✗ | ✓ | ✗ |
| 10X Multiome | ✓ | ✓ | ✓ | ✗ |
| **scATAC-seq module** | | | | |
| Single-cell ATAC-seq analysis | ✓ | ✗ | ✗ | ✗ |
| **Spatial transcriptomics module** | | | | |
| Clustering | ✓ | ✓ | ✓ | ✓ |
| Deconvolution | ✓ | ✓ | ✗ | ✗ |
| Xenium Analysis | ✓ | ✓ | ✓ | ✗ |
| Free | ✓ | ✓ | ✗ | ✗ |

'✓' and '✗' indicate the presence or absence of a functionality in the web server.

We next annotated the clusters with our in-house cell identification algorithm, CELLiD (https://www.immunesinglecell.org/cellpredictor), using the 'blood' cell type reference and CellTypist (https://www.celltypist.org/). CELLiD assigned the following labels, Memory CD4 T, CD14 monocyte, Memory B, Naive CD4 T, Naive B, CD16 monocyte, GZMB CD8 T, CD16 NK, GZMK CD8 T, Dendritic cell, and Megakaryocyte (Fig. 2C and Supplementary Fig. 3B). Using CellTypist, we identified 11 cell types, namely MAIT cells, B cells, Tcm/naïve helper T cells, non-classical monocytes, CD16 NK cells, Tem/Trm cytotoxic T cells, Tem/Effector helper T cells, classical monocytes, Tcm/naïve cytotoxic T cells, DC, and megakaryocytes/platelets. Both cell type annotation methods recapitulated the original annotations provided in the Seurat vignette. We note that the B cells could be split into naïve and memory B subsets, and CD8 T cells into GZMK and GZMB CD8 T cell subsets, providing a higher resolution of cell type labeling. We then confirmed the correctness of our annotation by checking marker gene expression (Supplementary Fig. 3F, G). ezSingleCell also provides the utility for users to rename clusters according to their preferences or to merge different clusters. Furthermore, users can subdivide a cluster of interest according to user defined parameters, which would aid in defining cell type subsets. We then computed the differentially expressed genes and found markers such as *CD79A*, *CD79B*, and *MS4A1* for the B cell subsets, and *S100A8*, *S100A9*, and *LYZ* for CD14 monocytes (Fig. 2D). Visualization of these genes using violin plots, feature plots, and ridgeline plots confirmed their higher expression in the respective cell type clusters (Supplementary Fig. 3C–E). Users can also compute for DEGs between two specified cell types and visualize the DEGs using a volcano plot with the top significant genes with the highest differential expression highlighted (Fig. 2E). For example, we observed *FCGR3A*, *S100A8*, and *LYZ* to be differentially upregulated in CD16 monocytes compared to CD14 monocytes.

We then computed cell type similarity (Fig. 3A), which showed consistency with the clusters observed in the UMAP plot. The cell types in the largest cluster (consisting mainly of T and NK cells) showed

## A. scRNAseq module

**Input data (raw counts / 10x cell ranger output / processed Seurat object)**

## B. UMAP and clustering

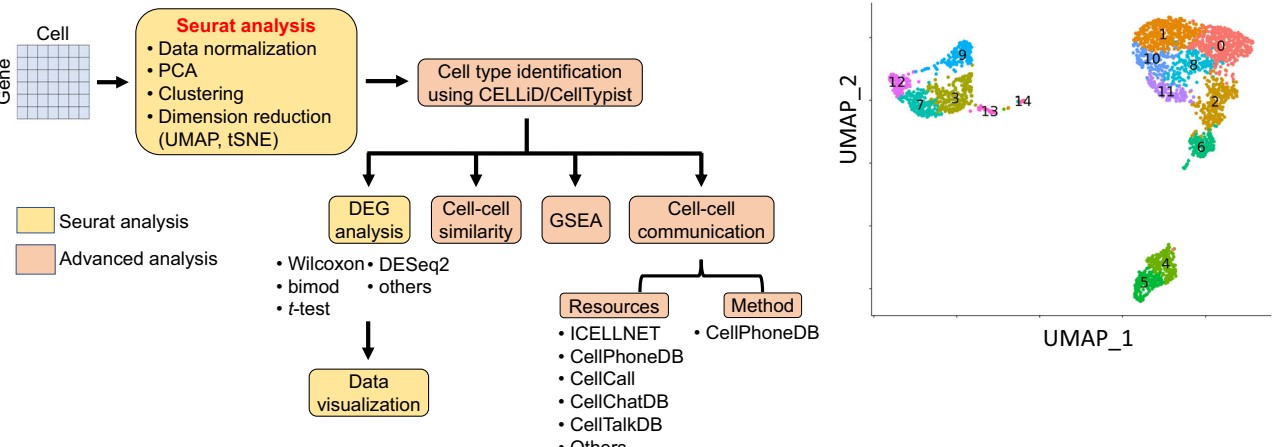

## C. Cell type identification and renaming

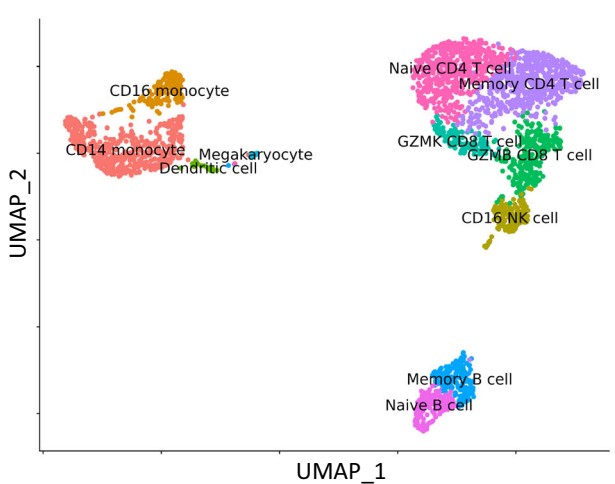

| Original ID | New ID |
|---|---|
| Memory CD4 T cell | Memory CD4 T cell |
| Naïve CD4 T cell | Naïve CD4 T cell |
| GZMB CD8 T cell | GZMB CD8 T cell |
| CD14 monocyte | CD14 monocyte |
| Memory B cell | Memory B cell |
| Naïve B cell | Naïve B cell |
| CD16 NK cell | CD16 NK cell |
| CD14 monocyte | CD14 monocyte |
| Memory CD4 T cell | Memory CD4 T cell |
| CD16 monocyte | CD16 monocyte |
| Naïve CD4 T cell | Naïve CD4 T cell |
| GZMK CD8 T cell | GZMK CD8 T cell |
| CD14 monocyte | CD14 monocyte |

## D. Cluster DEGs

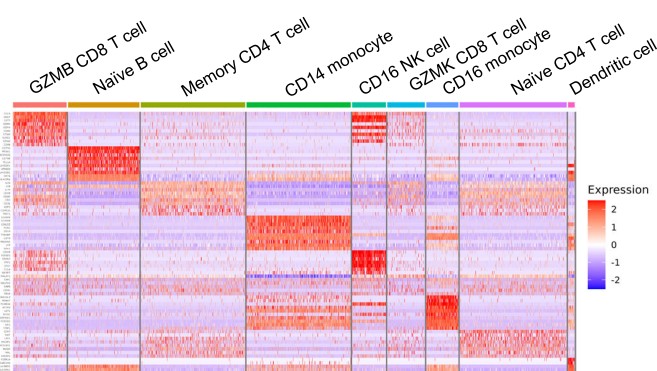

## E. Pairwise DEGs

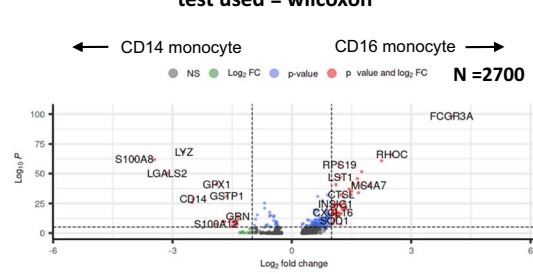

**Fig. 2 | ezSingleCell scRNA-seq module. A** Workflow of scRNA-seq analysis; (**B**) scRNA-seq UMAP and clustering visualization in ezSingleCell; (**C**) cell type identification using CELLiD and CellTypist. Users can also rename clusters in ezSingleCell; (**D**) Cluster-wise Differentially Expressed Gene (DEG) analysis using the 'wilcoxon' test; (**E**) Pairwise DEG analysis between two cell types of interest using the 'wilcoxon' test. Source data is provided as a Source Data file.

higher similarity score with other cell types in the same cluster, whereas isolated clusters (such as megakaryocyte) showed low similarity scores with all other cell types (Fig. 2C). This analysis of cell type similarity is useful when users identify a population of unknown cells and wish to estimate its similarity to known cell types.

ezSingleCell also offer fgsea[25] for gene set enrichment analysis (Fig. 3B). Users can choose any gene set from the MSigDB database for human (Hallmark, C1-C8) and mouse (MH, M1, M2, M3, M5, and M8) samples. Here we applied GSEA to the DEGs between the Naïve CD4 T and Memory CD4 T cells using the Human 'C7' immunologic

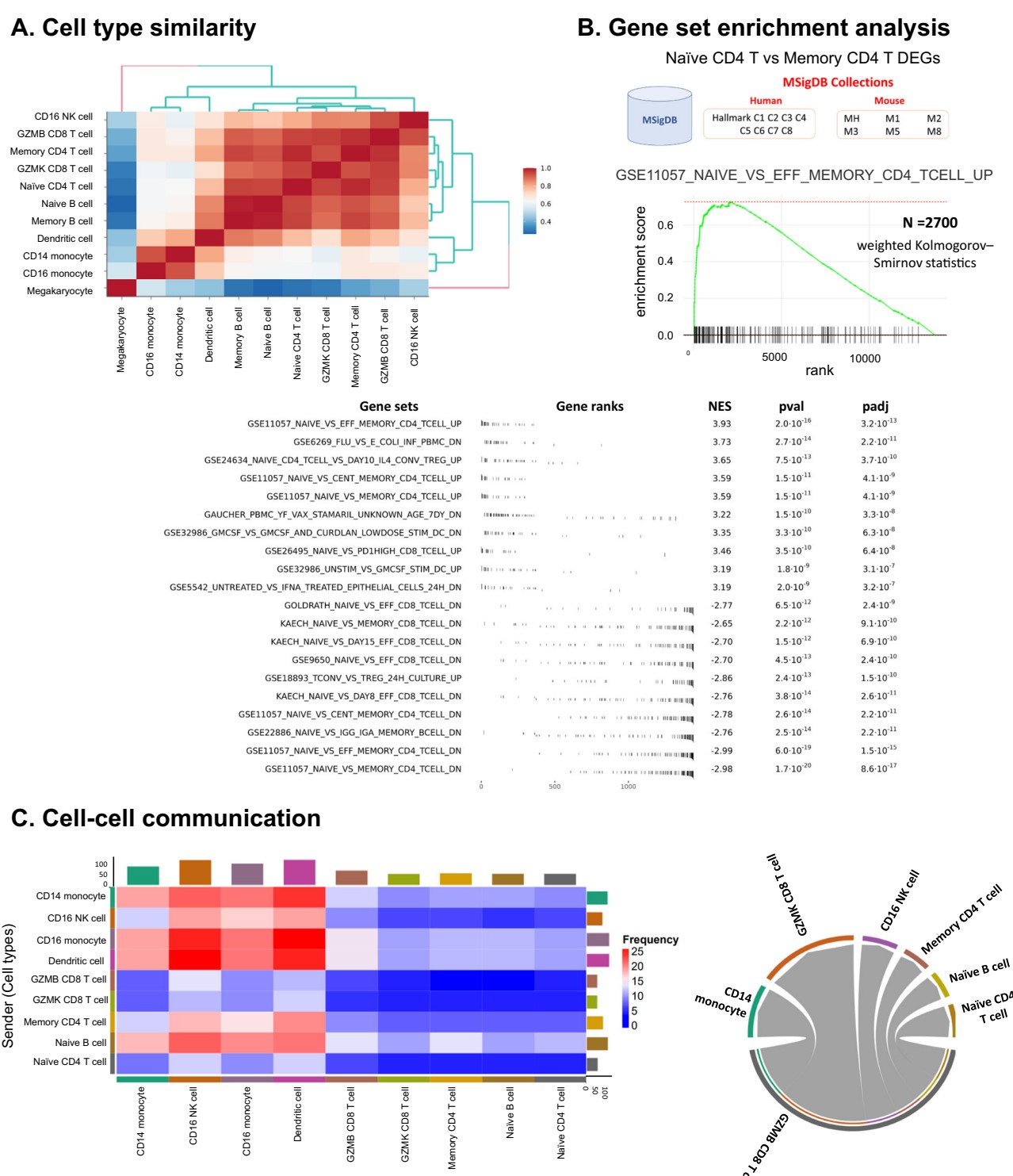

**Fig. 3 | Advanced analyses in ezSingleCell scRNA-seq module. A** Cell type similarity analysis; (**B**) Gene Set Enrichment Analysis (GSEA) using the weighted Kolmogorov–Smirnov statistic; (**C**) Cell-cell communication analysis using CellphoneDB. Source data is provided as a Source Data file.

signature gene set. We ranked the genes based on their log fold change and computed the top enriched pathways. As expected, we observed that the GSE11057 NAÏVE VS MEMORY CD4 TCELL UP, GSE11057 NAIVE VS EFF MEMORY CD4 TCELL UP and GSE11057 NAIVE VS CENT MEMORY CD4 TCELL UP gene sets were over-represented in Naïve CD4 when compared to Memory CD4. Other gene sets that also showed significant differences are listed in Fig. 3B. Lastly, ezSingleCell incorporates the widely adopted CellPhoneDB

package and various databases of ligand-receptor pairs for analysing cell-cell communication. Using CellPhoneDB, the cDC and CD16 monocytes were predicted to have the highest number of potential interactions (Fig. 3C left). We then filtered for significant ligand-receptor pairs between cDCs and CD16 monocytes using a *p* value cut-off <0.05 (Fig. 3C right).

Finally, the annotated output of the scRNA-seq module can be used in other modules. For example, users can navigate to the spatial

transcriptomics module and use the single-cell annotation to perform cell type/phenotype deconvolution.

## ezSingleCell's scIntegration module performs batch correction of multiple scRNA-seq datasets

To demonstrate batch effect correction of scRNA-seq data in ezSingleCell, we used a human PBMC dataset comprised of 2 batches. Both datasets were obtained from 10x Genomics, with the 3' batch data consisting of 8381 cells and the 5' batch data containing 7726 cells (Supplementary Table 3; Supplementary Dataset 1). The cell type annotations were retrieved from previous studies[26,27]. For batch effect correction, ezSingleCell takes in both the expression datasets and metadata with batch information. Cell type information is optional but required for cell type separation assessment downstream. We first performed the standard quality control and data pre-processing steps of log-normalization, finding the top 2000 highly variable genes, and scaling using the default parameters. Clustering was performed with the first 10 PCs, a $k$-nearest neighbor value of 10, and Louvain clustering at a resolution of 0.6. In the UMAP plot before batch correction, we can easily observe the batch effects present (Fig. 4). Four batch correction methods are currently available in ezSingleCell, namely Seurat, Harmony, scVI, and fastMNN. We ran the four methods with 2000 integration features and performed clustering and UMAP with the default parameters to visualize their results. All methods were able to successfully remove the batch effects (Fig. 4). To benchmark the batch mixing, we computed the median iLISI scores in ezSingleCell. The iLISI metric measures the number of batches within a local area and thus a score matching the number of batches indicate good mixing. Harmony ranked first (iLISI = 1.70), followed by fastMNN (1.54), Seurat (1.43), and scVI (1.39) (Fig. 4). Following batch integration, other downstream analysis such as clustering, cell type identification using the CELLiD algorithm, differential gene expression analysis, and marker gene visualization can be performed (Supplementary Fig. 4).

Like the scRNA-seq module, the annotated output of the scIntegration module can be used for cell type/phenotype deconvolution in the spatial transcriptomics module. Upon completion of analysis, users can navigate to the spatial transcriptomics module and employ the single-cell output.

## ezSingleCell's ST module performs clustering, integration, and deconvolution of spatial transcriptomics

In this module, we incorporated Seurat and GraphST for processing and analysis of spatial transcriptomics data. Seurat offers the key functionalities for both data pre-processing and clustering while GraphST performs spatially informed spatial clustering, multi-sample integration, and cell type deconvolution. GraphST is a state-of-the-art graph self-supervised contrastive learning method that achieved top performance in benchmarks against competing methods. The ST module can be used to analyse data acquired from different platforms such as Visium and Xenium from 10x Genomics with the latter offering sub-cellular resolution. Similar to the other modules, users can compute for cluster DEGs (Supplementary Fig. 5D) and pairwise comparison DEGs (Supplementary Fig. 5E) using a variety of statistical tests, and gene set enrichment analysis of spatial data with the 'fgsea' package (Supplementary Fig. 5F).

In this demonstration, we analysed a mouse brain sagittal anterior dataset acquired with 10x Genomics Visium and processed with the Space Ranger pipeline v1.1.0 (Supplementary Table 3; Supplementary Dataset 1). We first performed quality control, followed by normalization using SCTransform, selection of top 2000 highly variable genes, and data scaling. Users can first inspect the gene expression values on the tissue slide (Fig. 5A). Next, we performed PCA using the default parameters. For spatial clustering, we applied both Seurat and GraphST. We used the first 10 PCs, $k$-nearest

neighbour value of 10, Louvain clustering resolution of 0.6, followed by dimension reduction with UMAP to visualize the 15 clusters found (Fig. 5B). We also set the number of clusters to 15 for GraphST and observed that the clustering from GraphST was more consistent with the manual annotation (Fig. 5B).

ezSingleCell also offers the capability to deconvolute or annotate cell types in spatial spots using an annotated single-cell reference. For this purpose, ezSingleCell offers inter-module operability where users can navigate to the single-cell module, load the scRNA-seq dataset, perform data processing and annotation, before returning to the ST module for cell type/phenotype deconvolution. Users can employ the 'label transfer' function from Seurat or 'project_cell_to_spot' function from GraphST to perform deconvolution, and the results can be interactively visualized (Fig. 5C).

ezSingleCell supports analysis of sub-cellular resolution data such as those from the Xenium platform. Users can perform clustering analysis and interactively visualize the expression patterns at the sub-cellular level. In addition, users can zoom in to examine cellular composition and potential inter-cellular interactions. Users can also view the expression profile of each gene on the tissue slice (Fig. 5D).

## ezSingleCell's scMultiomics module performs joint analysis of multiple modalities

We next demonstrate ezSingleCell for multi-modal single-cell analysis with healthy PBMC datasets. The datasets consist of gene expression and protein expression (CITE-seq) data, as well as gene expression and chromatin accessibility (MultiOme) data. Both datasets were downloaded from 10x Genomics (Supplementary Table 3; Supplementary Dataset 1), where the CITE-seq dataset has 7865 cells and the Multi-Ome dataset comprised of 3012 cells. The CITE-seq dataset was pre-processed using the standard log normalization with default parameters and the second MultiOme with SCTransform normalization. Cell clustering was then performed using the first 10 PC dimensions, $k$-nearest neighbor value of 10, and clustering resolution of 0.6 (Fig. 6). For joint modality analysis of the CITE-seq data, Seurat WNN, a weighted combination of RNA and protein similarities, was used to calculate the KNN graph for clustering. Clustering at a resolution of 1.0 yielded 20 clusters for both individual modality and joint analysis. With CELLiD, we identified 16 cell types, namely CD14 monocyte, CD16 monocyte, CD16 NK cell, Cycling T/NK cell, Dendritic cell, GZMB CD8 T cell, GZMK CD8 T cell, MAIT cell, Memory CD4 T cell, Naïve B cell, Naïve CD4 T cell, Naïve CD8 T cell, pDC, Plasma cell, and Treg cell (Fig. 6). We next visualized the relevant markers such as MS4A1 in the RNA assay for B cells and CD4 in the ADT assay to verify the cell type annotations (Fig. 6). We also used DEG analysis to further verify the cell types by checking the top differentially expressed genes of each cluster.

We reran the same analysis with MOFA+ and clustered the data by specifying the number of clusters ($k$ parameter) to 10 and using all the latent factors from MOFA + . Using CELLiD, we identified 10 cell types, namely CD14 monocyte, CD16 NK cell, Cycling T/NK cell, Dendritic cell, MAIT cell, Megakaryocyte, Memory CD4 T cell, Naïve B cell, Naïve CD4 T cell, and pDC. We again visualized the marker genes such as MS4A1 for B cells and CD4 for CD4 T cells in both RNA and ADT assays to verify the cell type annotations (Fig. 6).

## ezSingleCell's scATAC-Seq module performs scATAC-seq data analysis and integration

In this final demonstration, we employed ezSingleCell to process an scATAC-seq dataset of 10k PBMCs from a healthy donor (Fig. 7A). The output from Cell Ranger ATAC was downloaded from 10x Genomics (Supplementary Table 3; Supplementary Dataset 1). For quality control, the user can compute metrics such as nucleosome banding pattern, transcriptional start site (TSS) enrichment score,

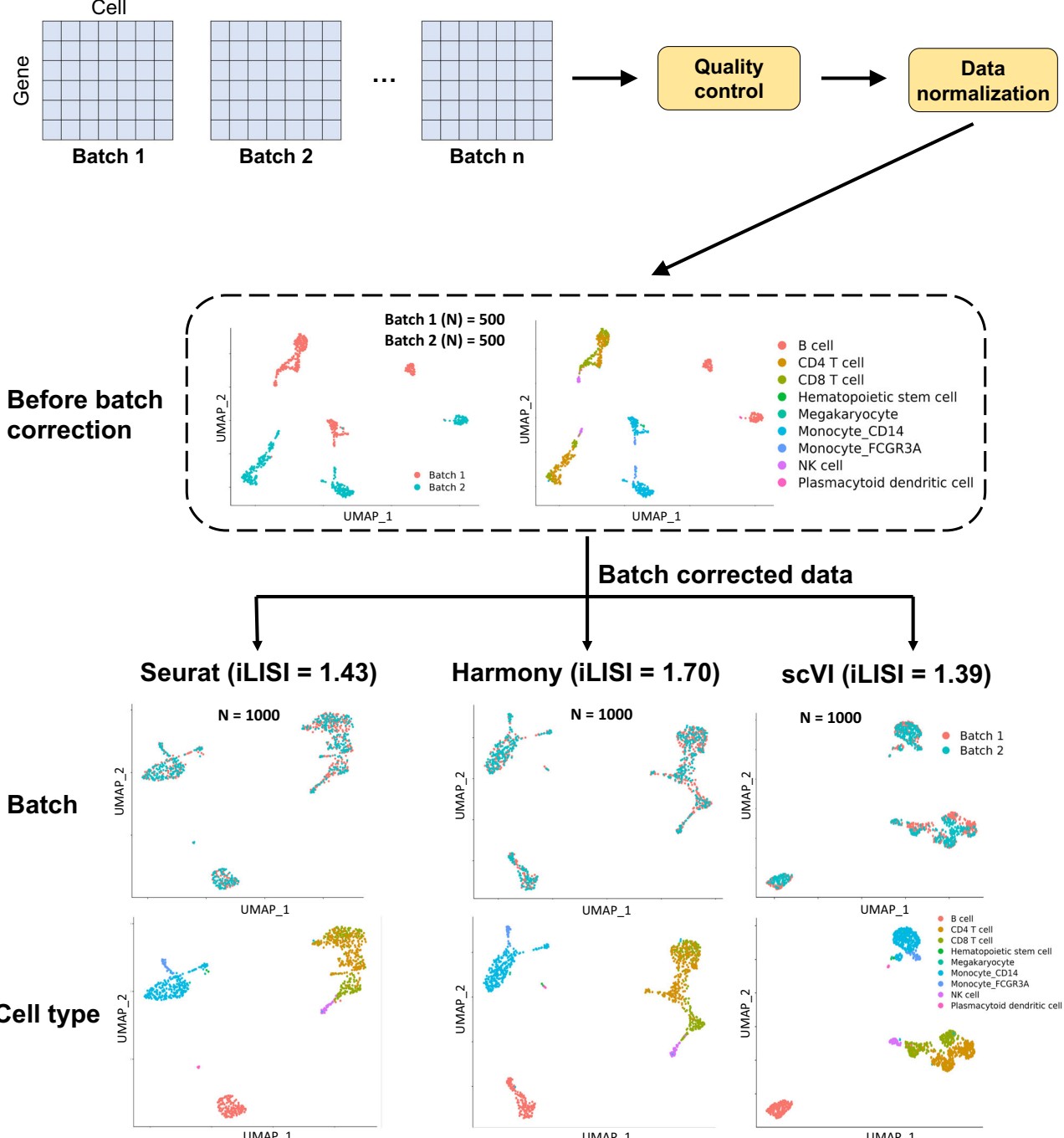

**Fig. 4 | ezSingleCell scIntegration module.** Major functionalities of this module include quality control, normalization, UMAP visualization before batch effect correction and after batch correction with Seurat, Harmony, or scVI, and iLISI scoring for integration assessment. A higher iLISI score indicates better batch mixing and performance. Source data is provided as a Source Data file.

total number of fragments in peaks, and ratio reads in genomic blacklist regions (Fig. 7B). We first normalized the data using Term Frequency-Inverse Document Frequency (TF-IDF) normalization, followed by feature selection and dimension reduction. Cell clustering was then performed using the first 10 PC dimensions with a *k*-nearest neighbor value of 10 and a Louvain clustering resolution of 0.6. The UMAP was then computed to visualize the 14 clusters found (Fig. 7C). We next performed differential peak analysis between clusters. The results were then visualized using violin plots,

feature plots, and coverage plots (Fig. 7D, E). Users can also compute for differentially expressed peaks (DE peaks) between clusters using a wide range of statistical tests (Fig. 7G).

ezSingleCell also offers inter-module operability between the scATAC-seq and scRNA-seq modules. Users can navigate to the scRNA-seq module, load, and process an scRNA-seq dataset, before returning to the scATAC-seq module to perform cell type label transfer for cell type annotation. In this example, we loaded a processed scRNA-seq dataset for human PBMCs and used it to identify 12 cell types in the

## A. Spatial data preprocessing

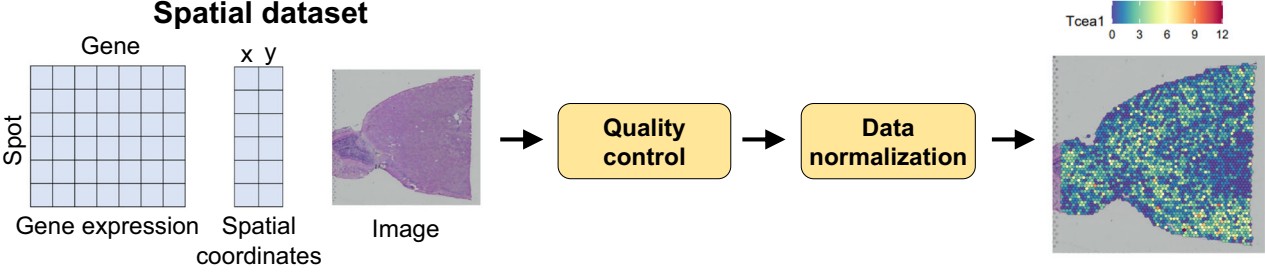

## B. Spatial clustering

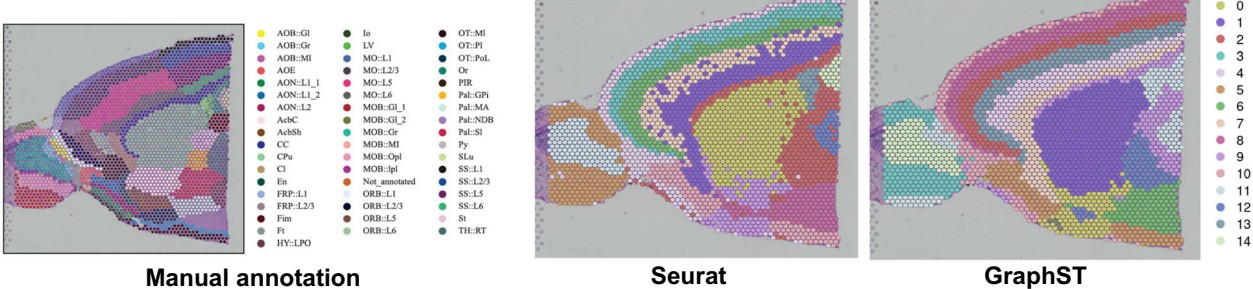

## C. Deconvolution

## D. Subcellular analysis

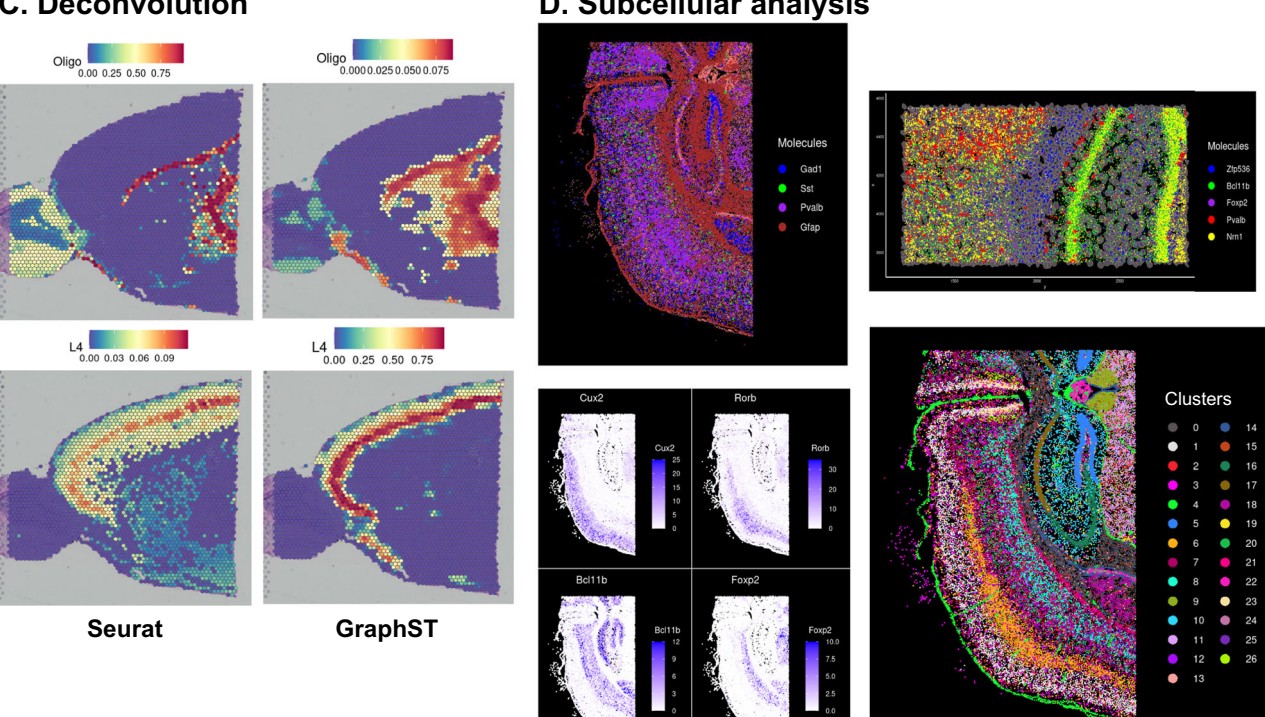

**Fig. 5 | ezSingleCell spatial transcriptomics module. A** Data input and pre-processing; (**B**) spatial clustering using Seurat and GraphST along with comparison with manual cell type annotation from pathologists; (**C**) cell type deconvolution using Seurat and GraphST showing the proportion of cell types deconvoluted with scRNA-seq reference data; (**D**) subcellular data (Xenium) analysis showing the clustering of molecules, visualization of expression profiles of molecules, and a zoomed in view of cell segmentation boundaries and individual molecules. Source data is provided as a Source Data file.

scATAC-seq data, namely CD4 Naïve cell, CD4 Memory cell, CD8 Naïve cell, CD8 effector cell, Double negative T cell, NK cell, pre-B cell, B cell progenitor, pDC, Dendritic cell, CD14+ Monocytes, and CD16+ Monocytes (Fig. 7H).

Users can link peaks to genes and visualize the signals in each cluster for a specific gene of interest (Fig. 7F) and perform gene set enrichment analysis using rGREAT[28] or fgsea[25] (Fig. 7I). In ezSingle-Cell, rGREAT (Genomic Regions Enrichment of Annotations Tool)

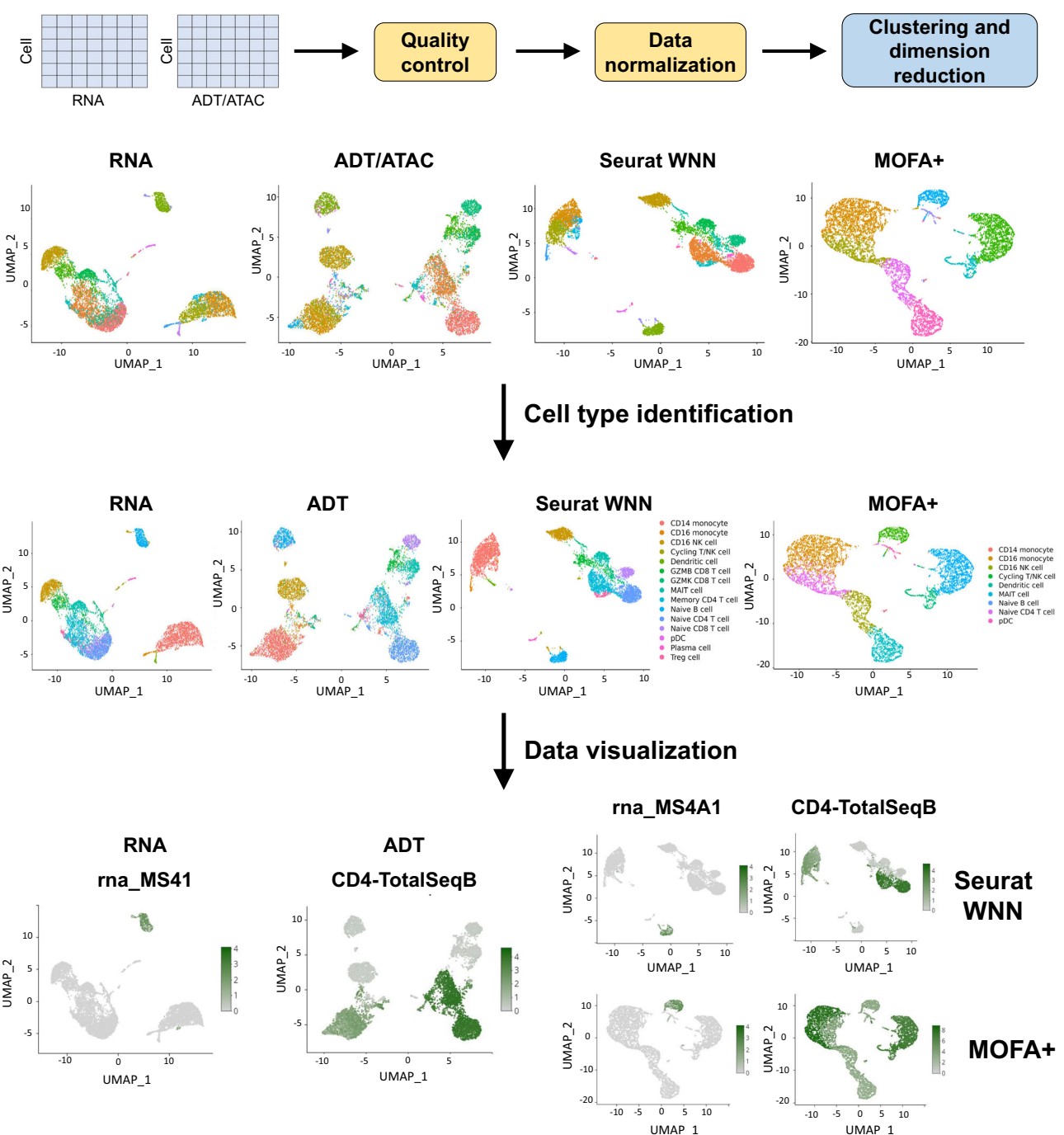

**Fig. 6 | ezSingleCell scMultiomics module.** The workflow includes data quality control, pre-processing, clustering, dimension reduction, cross-omics integration, post-integration analysis, and visualization. Currently, Seurat Weighted Nearest Neighbors (WNN) and MOFA+ are provided for cross-omics integration. After integration, cell types can be identified with the RNA modality and the joint clustering of Seurat WNN or MOFA + .Users can visualize specific genes and proteins in the joint UMAP. Here, we visualized the expression levels of B cell marker gene *MS4A1* and CD4 T cell protein marker CD4 in the joint UMAP visualizations from Seurat WNN and MOFA+ . Source data is provided as a Source Data file.

supports two species, human and mouse, with a variety of gene set collections and different TSS annotations such as txdb:hg19, TxDb.Hsapiens.UCSC.hg19.knownGene, RefSeq:hg19, GREAT:hg19, and Gencode_v19. rGREAT uses the differentially expressed genomic regions (or peaks) between two cell types as input and associates a

biological function with that region. As an example, we computed the DEGs between Naïve CD4 T and Memory CD4 T cells and performed gene set enrichment analysis using rGREAT and observed that the GSE11057 NAÏVE CD4 VS PBMC CD4 TCELL UP gene set was upregulated in Naïve CD4 T cell when compared to Memory CD4 T

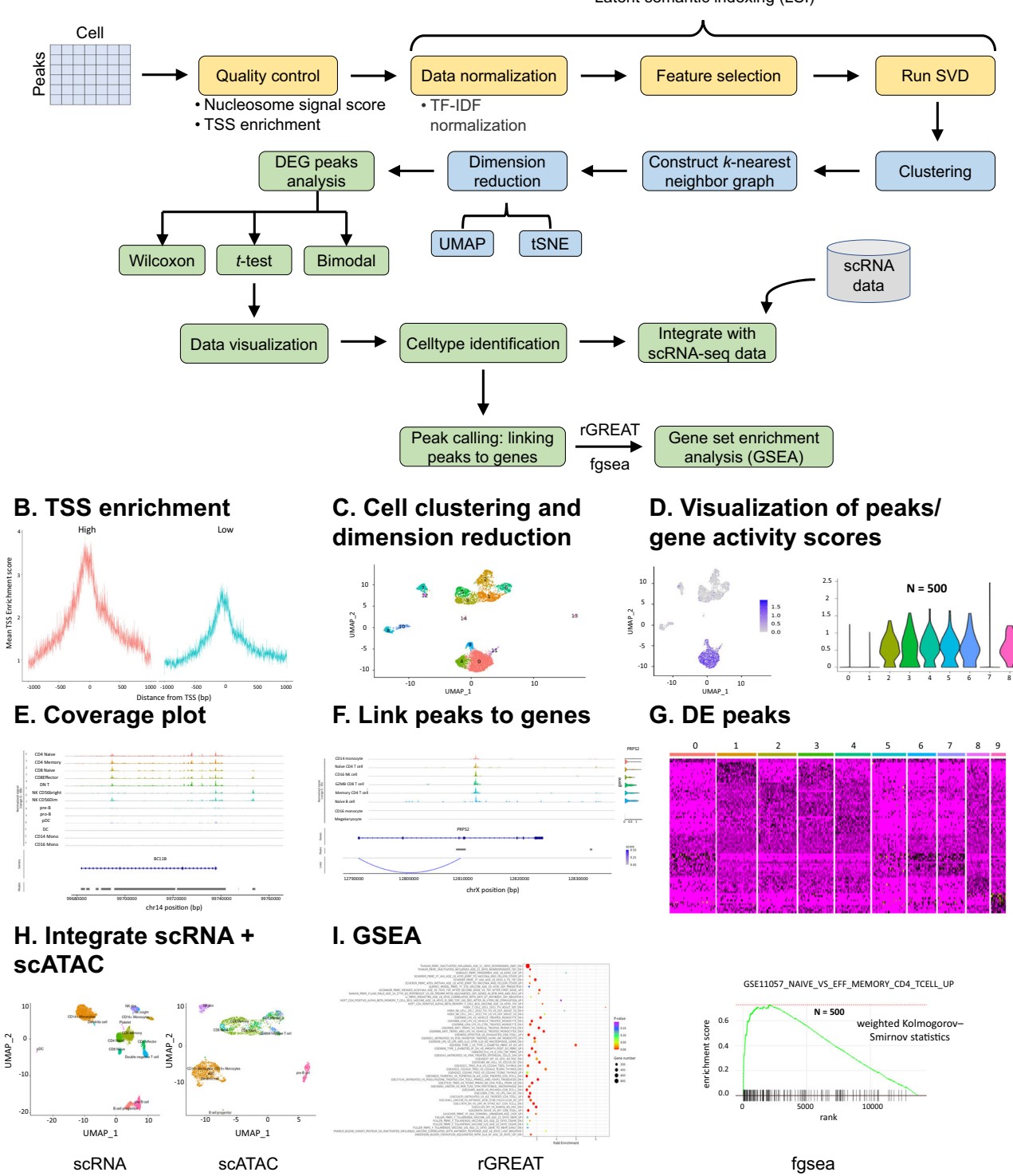

**Fig. 7 | ezSingleCell scATAC-seq module. A** Workflow of scATAC-seq analysis; (**B**) transcriptional start site (TSS) enrichment; (**C**) data clustering and dimension reduction; (**D**) data visualization; (**E**) coverage plot; (**F**) link peaks to genes; (**G**) differentially expressed peak (DE peak) analysis between clusters; (**H**) integration of scRNA-seq and scATAC-seq data for cell type identification. Here, we loaded a processed human PBMCs scRNA-seq dataset and identified 12 cell types in the scATAC-seq dataset through cell type label transfer; (**I**) gene set enrichment analysis (GSEA) of scATAC-seq data using the rGREAT and fgsea packages. We used the weighted Kolmogorov–Smirnov statistic for GSEA analysis. Source data is provided as a Source Data file.

cell. Like the scRNA-seq module, the scATAC-seq module also offers the fgsea package for gene set enrichment analysis where users can choose any human or mouse gene set from the MSigDB database. For example, we computed the pairwise DEGs between the Naïve CD4 T and Memory CD4 T cells using the gene activity matrix, and performed GSEA analysis with the 'fgsea' function and the Human 'C7' immunologic signature gene set to obtain pathways which were over-represented in Naïve CD4 compared to Memory CD4.

## Cross-module interaction in ezSingleCell

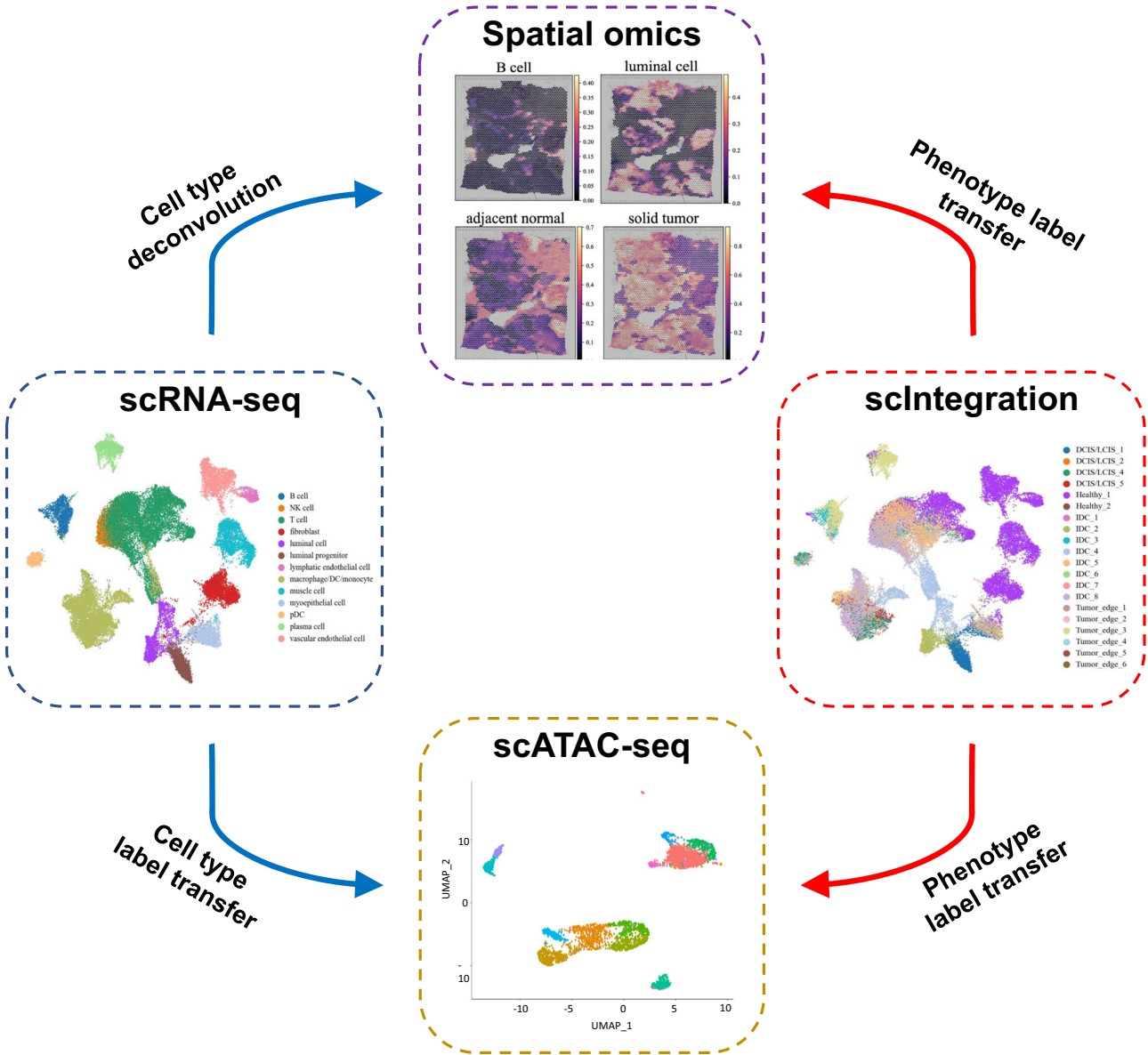

**Fig. 8 | ezSingleCell cross-module interaction capabilities.** Users can process single-cell RNA-seq data and use individual sets or batch integrated data to deconvolute cell types in spatial omics data or perform label transfer to accomplish cell type annotation of scATAC-seq data. Source data is provided as a Source Data file.

### ezSingleCell allows interplay of different modules

ezSingleCell allows inter-module operability wherein the user can perform data analysis in one module and use the results obtained in another module. To demonstrate this utility, we analysed a human breast cancer dataset acquired with 10x Genomics Visium and processed with the Space Ranger pipeline v1.1.0 (Supplementary Table 3; Supplementary Dataset 1). ezSingleCell currently implements four types of interactions (Fig. 8). In the scRNA-seq module at the step of cell type identification, users can click (1) the "Go to Spatial Deconvolution" button or (2) the "Annotate cell types for ATAC-data" button. The first option will lead them to the deconvolution step of the Spatial Transcriptomics module. Here the annotated scRNA-seq data will be used as the reference to deconvolute the cell type proportions in the spatial data using Seurat or GraphST. The second option will take the user to the

cell type identification step of the scATAC-seq module for cell type label transfer onto the scATAC-seq data with Signac. The labeled data in the scIntegration module can likewise be transferred. Furthermore, we also enabled two-way interactions. In the Spatial Transcriptomics module's deconvolution step, users can click on the "Load and process user reference dataset" button that will take the user to the scRNA-seq module to upload, analyse, and annotate their reference scRNA-seq data. The resulting annotated data will then be usable in the Spatial Transcriptomics module for cell type deconvolution. Similarly, the "Load and process user reference dataset" button at the cell type identification step of the scATAC-seq module navigates to the scRNA-seq module for scRNA-seq data processing and analysis. The processed scRNA-seq data is then made available to the scATAC-seq module for cell label transfer.

### ezSingleCell can handle large datasets without compromising performance

To handle large datasets, ezSingleCell employs a technique named 'geometric sketching' to subsample large scRNA-seq datasets while preserving rare cell types and cell states. By employing this feature in ezSingleCell, users can perform data analysis of large datasets with accelerated clustering, visualization, and integration analyses. In a user test scenario of data comprising 50,000 cells, users can perform basic and advanced single-cell data analysis composed of clustering, dimension reduction, and cell type identification within 5–6 min, and within 15 min for 100,000 cells.

## Discussion

Single-cell profiling of cells is generating huge amounts of data. Despite the increasing availability of analytical tools to exploit the data, data analysis by bench scientists is hampered by minimum bioinformatics skill requirements of these tools. ezSingleCell is an integrated one-stop single-cell and spatial omics analysis platform with an intuitive GUI designed for users with no bioinformatics background. ezSingleCell has analysis modules to cover the data generated by different single-cell omics experiments, namely scRNA-seq, scIntegration, scMultiomics (CITE-seq, 10x Multiome), scATAC-Seq, and spatial omics. ezSingleCell accomplishes this by combining in-house novel algorithms such as CELLiD for cell type identification and GraphST for spatial clustering and deconvolution, along with other established algorithms for both basic and advanced analyses such as batch effect correction, gene set enrichment analysis, cell-cell communication, and spatial deconvolution. The GUI is designed to be user-friendly for interactive data exploration and analysis. Users can customize their data analysis with a wide range of parameters. ezSingleCell also accepts inputs in different formats such as text files or Cell Ranger / Space Ranger/ Cell Ranger-ATAC output and produces publication ready figures and tables. ezSingleCell is available in two forms: an installation-free web application (https://immunesinglecell.org/ezsc/), or a software package with a Shiny app interface (https://github.com/JinmiaoChenLab/ezSingleCell2) that can be run on a computer for offline analysis. ezSingleCell's source code is also available on Zenodo (https://doi.org/10.5281/zenodo.10785313).

In the future, we will continue to maintain and upgrade ezSingleCell. As the number of cells profiled per experiment is increasing rapidly, this results in very large datasets being generated and consequently analysis and integration of such big data is time consuming and memory intensive. In the future versions of ezSingleCell, we will incorporate novel deep learning methods for more efficient dimension reduction, clustering, and batch integration. We will also extend the current spatial transcriptomics module to analyze spatial proteomics and spatial multi-omics. We will also add functionalities for inferring of cell-cell interactions based on spatial proximity and the expression of ligand-receptor pairs. With the emergence of the latest sub-cellular spatial technologies such as Nanostring CosMX, Stereo-Seq, Vizgen MERSCOPE, PixelSeq, and SeqScope, we plan to incorporate novel algorithms for better cell segmentation to process such datasets.

## Methods

### Data input

ezSingleCell accepts input from multiple technologies (Smart-Seq2, 10x, CITE-Seq, Multiome, and Visium) in a variety of formats: (i) text files (txt, csv, or tsv) or 10x Cell Ranger output for scRNA-seq and data integration analysis, (ii) 10x CITE-Seq count output for CITE-Seq, and 10x Cell Ranger ARC output for multiome, (iii) 10x Cell Ranger ATAC output for scATAC-Seq analysis, and (iv) 10x Space Ranger output for spatial transcriptomics (Visium) and Xenium output for Xenium data analysis.

### Data pre-processing

For scRNA-seq data, ezSingleCell offers Seurat's functions for quality control and data normalization with log-normalization or SCTransform[29]. The normalized data can then be scaled and dimensionally reduced with Principal Component Analysis (PCA) (Fig. 2A). For the scIntegration module for single-cell data integration, the pre-processing steps are the same as those of the scRNA-seq module. For sc-multi-omics data, ezSingleCell has two methods available for multimodal analysis: Seurat WNN[9] and MOFA + [19]. For Seurat WNN, both assays (RNA and ADT for CITE-seq data; RNA and ATAC for 10x scMultiOme data) are first independently pre-processed and dimensional reduction is performed. The closest neighbors are thereafter calculated for each cell based on a weighted combination of RNA and protein similarities. The resulting WNN is usable for clustering and visualization. For MOFA + , the initial pre-processing steps such as data normalization and scaling for each modality are performed separately. After pre-processing, MOFA+ infers a low-dimensional representation of the data in terms of a small number of (latent) factors that capture the global sources of variability. MOFA+ employs Automatic Relevance Determination (ARD), a hierarchical prior structure that facilitates untangling variation that is shared across multiple modalities from the variability present in individual modalities (Fig. 6).

For scATAC-Seq data, ezSingleCell employs the Signac package[20] for term frequency-inverse document frequency (TF-IDF) normalization. This is a two-step normalization procedure that both normalizes across cells to correct for differences in cellular sequencing depth and across peaks to give higher values to rarer peaks. This is followed by feature selection using only the top n% of features (peaks) for dimensional reduction or removing features present in less than a specified number of cells. Singular value decomposition (SVD) is then performed on the TD-IDF matrix of the selected features (peaks) to return a reduced dimension representation. This workflow is known as latent semantic indexing (LSI). Users can visualize the correlation between each LSI component and sequencing depth (Fig. 7A).

For spatial transcriptomics data, quality control functions are available for both 10x Visium and Xenium data, and normalization is done using SCTransform[30]. In spatial data, the variance in molecular counts per spot can be substantial and is also dependent on the nature of the tissue sample, particularly if there are differences in cell density across the tissue. For example, tissues that are depleted for neurons (such as the cortical white matter), reproducibly exhibit lower molecular counts. As a result, standard approaches such as log normalization which forces all cells or spots to have the same total count, are inappropriate. Therefore, SCTransform is recommended over log-normalization as it employs regularized negative binomial models of gene expression to account for technical artifacts while preserving biological variance (Fig. 5).

### Clustering and dimension reduction

For scRNA-seq data clustering, ezSingleCell employs functions from the Seurat package, which employ graph-based community detection where a k-nearest neighbor (KNN) graph is constructed based on the Euclidean distance in the PCA space and the edge weights between two cells are refined based on the shared overlap in their local neighborhoods (Jaccard similarity). Modularity optimization techniques such as the Louvain algorithm[31] are then applied to iteratively cluster the cells together to optimize the standard modularity function. The user can adjust the clustering resolution to obtain broad or fine-grained clusters with higher values leading to more clusters. For visualization, UMAP and tSNE dimensionality reductions are available and the user can specify the number of PCA dimensions to be used (Fig. 2B).

For sc-multi-omics data, the closest neighbors of each cell are calculated based on a weighted combination of RNA and protein similarities. With the Seurat analysis option, the cell-specific modality weights and multimodal neighbors are calculated using the

'FindMultiModalNeighbors' function. Clustering is then performed using a shared nearest neighbor (SNN) modularity optimization technique such as the Louvain algorithm. For MOFA + , data clustering is performed using the 'cluster_samples' function, specifying the number of clusters and the latent factors from the model trained after running the 'run_mofa' function (Fig. 6).

To cluster spatial data, ezSingleCell offers two methods, Seurat and GraphST. Seurat use a graph-based community detection algorithm such as Louvain or Leiden clustering. GraphST combines graph neural networks with self-supervised contrastive learning to learn latent representations of spots from gene expression and spatial information to accomplish spatially informed clustering. GraphST currently offers three clustering methods, 'mclust'[32], Leiden, and Louvain clustering. In our tests, we found that mclust performs better than Leiden and Louvain in most cases. Therefore, we recommend using mclust for this step (Fig. 5).

Finally, clustering for scATAC-seq data follows the same procedure as processing single cell RNA-seq data by employing graph-based clustering and non-linear dimension reduction. The functions 'RunUMAP', 'FindNeighbors', and 'FindClusters' from the Seurat package are available (Fig. 7).

## Cell type identification
To identify the cell type of clusters, ezSingleCell offers our in-house CELLiD algorithm and CellTypist. CELLiD employs the average expression of clusters for correlation analysis with reference datasets from the DISCO database (https://www.immunesinglecell.org/) to infer the cell types. It is available in the scRNA-seq (Fig. 2C), scIntegration (Supplementary Fig. 4), and scMultiomics (Fig. 6) modules. CellTypist is a cell type annotation tool for single cell data that uses logistic regression classifiers trained with the stochastic gradient descent algorithm. CellTypist can perform cell type prediction using either built-in (with a current focus on immune sub-populations) or a user customized reference.

## Cell type similarity
Cell type similarity of cell clusters is a metric to measure the closeness of a particular identified cluster to others. Users can choose to compute using cosine similarity, Pearson correlation, or Spearman correlation. This utility is available in all modules.

## Data visualization
ezSingleCell offers UMAP, tSNE, violin, feature, ridge, and volcano plotting functionalities for data visualization (Supplementary Fig. 3C–E). For spatial datasets, users can visualize their data with interactive plots showing the spatial localization of features measured (Supplementary Fig. 5A). For scATAC-seq datasets, users can visualize their data using the CoveragePlot function (Fig. 7E) that shows the transcription factors, genes, and peaks within given regions of the genome.

## Batch effect correction and data integration
ezSingleCell contains four popular and best-performing methods, Seurat, Harmony[16], scVI[17], and fastMNN[33] (Fig. 4), which were identified in previous benchmarking studies[27,34]. In this data integration step, users can customise the tuning parameters such as the number of features to integrate. The ARI, iLISI, and cLISI metrics are available to quantitatively benchmark the data integration results.

## scRNA-seq downstream analyses
Differentially expressed genes analysis (both cluster and pairwise) is available in ezSingleCell using the 'FindAllMarkers' and 'FindMarker' functions from the Seurat package with statistical tests like Wilcoxon, bimod, and *t* test available (Fig. 2D, E). Other available downstream analyses include gene set enrichment analysis (GSEA)[35]

analysis with gene sets from the MSigDB database (Fig. 3B) and cell-cell communication with CellphoneDB[18] (Fig. 3C). For cell-cell communication prediction, users can select from multiple reference resources such as ICELLNET, CellPhoneDB, CellChatDB, CellTalkDB, and OmniPath.

## Single-cell ATAC-seq module
For processing ATAC-Seq data, ezSingleCell uses MACS2 to perform peak calling. Users can visualize the results with a coverage plot (Fig. 7E) and perform a differential accessibility (DA) test to find differentially accessible regions between clusters using logistic regression (Fig. 7G). Users can link peaks to genes and visualize for a specific gene of interest (Fig. 7F), and perform gene set enrichment analysis similar to the scRNA-seq module using the rGREAT and fgsea packages (Fig. 7I). In gene set enrichment analysis using rGREAT, users can select from a large number of gene set collections (such as BP, CC, MP, H, C2, C5, C7, and C8) and different TSS annotations such as txdb:hg19, TxDb.Hsapiens.UCSC.hg19.knownGene, RefSeq:hg19, GREAT:hg19, and Gencode_v19. Thereafter, rGREAT takes the differentially expressed genomic regions (or peaks) between two cell types as input to identify biological functions associated with those regions. For fgsea, ezSingleCell offers gene sets from the MSigDB database for human (Hallmark, C1-C8) and mouse (MH, M1, M2, M3, M5, and M8) for analysis with the 'RNA' assay after estimating pairwise DEGs between two cell types of interest.

## Integrating scRNA-seq and scATAC-seq data
Users can integrate their scATAC-seq data with scRNA-seq data by performing cross-modality integration and label transfer. In the integration process, ezSingleCell estimates the scRNA-seq levels from scATAC-seq by quantifying gene expression 'activity' from the scATAC-seq reads. The 'internal' structure of the scATAC-seq data is then learned using LSI. The 'anchors' between the scATAC-seq and scRNA-seq datasets are then identified, followed by data transfer between datasets (either transfer labels for classification or impute RNA levels in the scATAC-seq data to enable co-embedding). Simuntaneously, the reference scRNA-seq data is either processed using the scRNA-seq module or an already processed scRNA-seq reference dataset can be loaded. Finally, the scRNA-seq and scATAC-seq cells are co-embedded in the same low dimensional space using the same anchors to transfer cell type labels to impute scRNA-seq values for cells in the scATAC-seq data. The measured and imputed scRNA-seq data are then merged and can be visualized with a UMAP plot (Fig. 7G).

## Integration of scRNA-seq and spatial data (deconvolution)
Users can employ two different methods, Seurat and GraphST, to integrate spatial data with scRNA-seq reference data to annotate or deconvolute the cell types present in the spatial data. Publicly available scRNA-seq references are available at the DISCO database (https://www.immunesinglecell.org/) or the Human Cell Atlas (https://www.singlecell.broadinstitute.org) for human data, and 10x Genomics Visium and Xenium spatial datasets of human and mouse samples can be obtained from the 10x Genomics website (https://www.10xgenomics.com/resources/datasets). The scRNA-seq module of ezSinglecell can be used to process and annotate user uploaded single-cell data, and the output will be directly usable to deconvolute cell types in the spatial data. For Seurat deconvolution, ezSingleCell uses the 'label transfer' function to output a probabilistic classification (or prediction scores) for each of the scRNA-seq cell type predicted to be present in each spot. These predictions can then be added back to the project's Seurat object. GraphST is a deep learning-based algorithm that can deconvolute cell types in spatial data using a high quality scRNA-seq reference data. GraphST first trains an auto-encoder to learn informative cellular features from the scRNA-seq data in an unsupervised way. Thereafter, GraphST learns a mapping matrix to project the

scRNA-seq data into the ST space based on their learned features via an augmentation-free contrastive learning mechanism where the similarities of spatially neighboring spots are maximized while those of spatially non-neighboring spots are minimized. The mapping matrix is then used to estimate the cell-type compositions of ST spots.

## Statistics and reproducibility

No statistical method was used to predetermine the sample size. The data was downloaded from 10x's public repository. No data were excluded from the analyses. The experiments were not randomized. The investigators were not blinded to allocation during experiments and outcome assessment. We use statistical tests such as "wilcoxon", "bimod", "roc", "t-test", "negbinom", "poisson", "LR", "MAST", "DESeq2" in some steps such as Differentially Expressed Genes (DEGs) analysis and Gene Set Enrichment Analysis (GSEA).

## Reporting summary

Further information on research design is available in the Nature Portfolio Reporting Summary linked to this article.

## Data availability

All relevant data used in this study has been listed in Supplementary Table 3 and Source Data File of the manuscript. The data used in this study has been uploaded to Zenodo and is freely available at https://zenodo.org/records/10609310. Source data is provided with this paper.

## Code availability

ezSingleCell is available in two forms: an installation-free web application (https://immunesinglecell.org/ezsc/) or a software package with a shinyApp interface (https://github.com/JinmiaoChenLab/ezSingleCell2) for offline analysis. ezSingleCell's source code is also available on Zenodo (https://doi.org/10.5281/zenodo.10785313)[36].

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

## Acknowledgements

We would like to acknowledge members of Chen Jinmiao's lab in SigN, A*STAR. This work was supported by A * STAR, AI, Analytics and Informatics (AI3) Horizontal Technology Programme Office (HTPO) seed grant C211118015, the Singapore National Medical Research Council Open Fund Individual Research Grant [NMRC-OFIRG18nov-2013], Singapore National Medical Research Council Open Fund Large Collaborative Grant [NMRC/OFLCG/003/2018], and the National Research Foundation Grant (NRF-CRP19-2017-04) to J.C.

## Author contributions

J.C. conceptualized and supervised the project, R.S. developed the ezSingleCell webserver, R.S., K.S.A., and J.C. wrote the manuscript, M.L. developed and helped to incorporate CELLiD, Y.L. developed and helped to incorporate GraphST, J.L. provided feedback to improve the webserver.

## Competing interests

The authors declare no competing interests.
