## [Peer Review File · Nature Communications]

ezSingleCell: An integrated one-stop single-cell and spatial omics analysis platform for bench scientistsReviewer #1 (Remarks to the Author):

Work from Raman Sethi and colleagues developed ezSingleCell, a web application and a shinyApp that integrates analysis tools for multiple single-cell and spatial omics data types into a pipeline designed to facilitate researchers with no programming background. The pipeline is quite comprehensive, encompassing not only the entire process from data pre-processing to downstream pathway analysis and cell-cell interactions but also processes such as heterogeneous data integration and cellular deconvolution of spatial-omics data. In addition, they provided a detailed help document to help users get started. While I appreciate the authors' efforts, I do not believe this manuscript can be published in its current form. There are several reasons for my decision:

- 1) It is not clear whether this work has sufficient contribution and innovation. The pipeline they developed, while comprehensive, is not innovative as it simply makes an interactive interface that calls on some existing packages and does not contribute anything new.
- 2) This tool offers a range of modules that are independent of each other (e.g., 'single cell RNA-Seq', 'single-cell data integration', 'single cell ATAC-seq', etc.) to enable analysis and integration of different experimental data. Since the modules are independent, this software looks like several software glued together. For the analysis that each module would like to perform, there are nowadays a lot of packages that perform similar, or even more detailed functions. Therefore, for the current version, it is hard to see the indispensability of this software. However, the unique status of the software might be demonstrated if the authors could go further to implement modules and function interaction (like what they mentioned in the discussion).
- 3) It is difficult to conclude whether this all-in-one software can really help the researchers' work and save their time. First of all, the software only called some of the functions of some popular single-cell analysis packages and fixed most of the adjustable parameters, meaning many functions carried by the original package are not able to use (for instance, it is not possible to specify only two cell groups for DGE analysis). Secondly, this integrated pipeline leads to the need to restart the program from the first step again if something goes wrong at an intermediate step. Thirdly, it is also impossible to modify or delete any clustering or prediction results, even if we found some of them, even if we found that some inferences may be not reasonable biologically.
- 4) With rapid advances in biotechnology and computational methods, there are dozens of algorithms that perform the same or similar functions. However, the authors did not declare why they chose to call the package specified in the article. For example, why use CELLiD to identify cell types, and why use CellphoneDB to compute cell interactions? Is this due to the considerations of time complexity, space complexity, or accuracy?
- 5) How could users evaluate the different clustering results obtained from different parameters in cell clustering analysis, and determine the optimal number of clusters?

In general, I encourage the authors to reflect on the innovation of this work and to put themselves in the users' shoes to design the software's architecture. Unfortunately, in its current state, I do not recommend accepting this manuscript.

Reviewer #2 (Remarks to the Author):

Sethi et al., presented an interactive tool for single-cell and spatial omics analysis, namely ezSingleCell. By assembling current state of art bioinformatic methods, it aims to enable bench scientist for hassle-free in-depth analysis. Compared to other tools with similar purposes, ezSingleCell offers more functionalities including cell-cell communication, spatial and scATAC-seq analysis. And this is facilitated by a web server and/or shinyApp software that runs on a PC. I find the tool may help bench scientists to run their single-cell and spatial data and have some initial explorations or even publication ready figures. But the current version appears not mature yet, and further effort and resources are required.

Major points:

1. It runs well when "Load example and run", but did not work when I uploaded local single-cell

data, for example a h5 file as stated in the manuscript, And it raised error " Error: argument is of length zero" (google chrome browser)

2. The server appears not stable as even loading of example data, running cellid function already slowdowned or crashed the server. Though it is understandable that such analysis may require high computational resource. With more than 3000 cells (often 100k cells for almost ready study cases nowadays) and concurrent users, and I think it can hardly work without massive infrastructures. I suggest the authors to specify what they can offer for this server, and what is their limit.

3. In the single-cell data integration "Quality control "section, no threshold setting option to do the filtering, so actually no quality control. This apply to some other functionalities, where few option is given.

4. Line 166 and supplementary Figure 3F: KLRB1 is not a good MAIT marker, instead, the author should use SLC4A10 etc. The MAIT annotation seems not correct not only because of the wrong marker used, but also the proportion was way higher than expected if the authors' annotation is used. In another word, the accuracy of the pipeline is questioned.

Minor points:

1. Line 109: table 4 appears before table 2 and 3

2. For the interface design, I noticed some pages showed multiple buttons, since its end users are researchers without any programming experience, I suggest the authors to put numbers on the buttons to have a clear order, which prevents ambiguous mistakes.

3. Line 167: Naive and memory B cell markers need to be specified in the figure. In addition, it seems not difficult to split naive and memory given the cell amount used in this particular dataset.

4. Line 229: figure 7F appears before 7E, such order mistakes appear several times.

5. Layout of some Figures should be carefully aligned or names. i.e. Figure 3 and 7F, as well as the legends, with a bit more information.

Reviewer #3 (Remarks to the Author):

Sethi et al. developed ezSingleCell, a convenient and easy web service for single cell and spatial analysis without requiring prior programming knowledge. Overall, this study could benefit researchers without bioinformatics expertise or with limited access to computing resources. Meanwhile, it could probably aid the study of single-cell and spatial omics. In general, I would like to make the following suggestions for improvement.

1. The authors should specify their plan to maintain and upgrade ezSingleCell. Quite a few websites are no longer timely and efficiently upgraded after publication. How can authors guarantee the efficient maintenance of ezSingleCell in five years.

2. Currently, a lot of online analytic tools (CancerSEA, SPEED, SPatialDB, Galaxy Single Cell Omics Workbench) have been developed. Authors are suggested to compare ezSingleCell with those already published tools, either in introduction or in discussion section, to highlight the uniqueness and novelty of ezSingleCell.

3. Authors should test the performance of ezSingleCell on larger data sets (cell number between 50, 000 and 100, 000).

4. I am curious about the performance of ezSingleCell on other scRNAseq techniques, for example sci-RNA-seq3.

5. Could the author integrate target prediction and GO/KEGG enrichment function into the scATAC-seq module of ezSingleCell?

6. How many species does ezSingleCell support? Could ezSingleCell be employed to analyze the data of other species (e.g., fruit fly, zebrafish, worm)? Have the authors tested what problems it might happen when dealing with scRNAseq data from other species. Do the authors have any plan to solve those problems?

Response to comments for paper NCOMMS-23-12483-T “ezSingleCell: An integrated one-stop single-cell and spatial omics analysis platform for bench scientists”

Firstly, we are grateful to the reviewers for their time spent reviewing the manuscript and testing the ezSingleCell software. The comments contain many critical and useful suggestions that helped us improve ezSingleCell. Based on the feedback given, we have extensively updated ezSingleCell. The changes implemented are as follows:

1. We have revised ezSingleCell to allow interplay between different modules, allowing users to employ the output of a module in the data analysis of another module. For example, the output of cell types identified from single cell RNA-sequencing module can be used to (i) deconvolute the cell type proportions in spatial transcriptomics data, and (ii) perform cell type label transfer in single cell ATAC-seq data. To our best knowledge, no other existing web servers have this functionality.
2. We have expanded ezSingleCell to incorporate more methods such as scVI for single-cell RNA-seq data integration, and CellTypist for automated cell type annotation. We are also working on incorporating totalVI for multi-omics analysis, and RCTD and cell2location for spatial transcriptomics deconvolution.
3. We have fixed various bugs that result in errors and prevent the smooth running of the data processing pipelines.
4. We have enabled large data analysis by incorporating the sketch package that implements geometric sketching. Geometric sketching is a method to subsample massive scRNA-seq datasets while preserving rare cell states. This is useful for accelerating clustering, visualization, and integration analyses. For example, basic single cell analysis comprising of variable feature selection, dimension reduction, clustering, and cell type identification for around 50,000 cells can be done in 6 minutes (excluding upload time) while analysis of 100,000 cells can be done in 15 minutes (excluding upload time).
5. We have now incorporated clustree (<https://github.com/lazappi/clustree>) to assist users in selecting a suitable number of clusters. Users can also merge clusters at the cell type annotation stage by renaming them as appropriately.
6. We also enable users to select a cluster of interest, and subset them according to their own choice of criteria.
7. We have moved ezSingleCell to a new hosting server with more compute resources to facilitate analysis of larger datasets by more users in parallel. We have upgraded the server with 2 GPUs (GeForce RTX 3060) to accelerate some analyses especially those that employ deep learning such as GraphST and scVI. Each GPU has 12 GB GDDR6 graphics memory with a 192 Bit memory bus.
8. We have added functionality for filtering cells at the QC stage for all 5 modules. For example, users can filter cells by mitochondrial percentage, minimum number of features, or filter features by minimum number of cells.

Reviewer #1 (Remarks to the Author):

Work from Raman Sethi and colleagues developed ezSingleCell, a web application and a shinyApp that integrates analysis tools for multiple single-cell and spatial omics data types into a pipeline designed to facilitate researchers with no programming background. The pipeline is quite comprehensive, encompassing not only the entire process from data pre-processing to downstream pathway analysis and cell-cell interactions but also processes such as heterogeneous data integration and cellular deconvolution of spatial-omics data. In addition, they provided a detailed help document to help users get started. While I appreciate the authors' efforts, I do not believe this manuscript can be published in its current form. There are several reasons for my decision:

*1) It is not clear whether this work has sufficient **contribution** and innovation. The pipeline they developed, while comprehensive, is not innovative as it simply makes an interactive interface that calls on some existing packages and does not contribute anything new.*

Response: We thank the reviewer for the important and relevant comments. The comments are useful in helping us focus on the shortfalls for improvement and clarification. We have carefully considered the comments and revised ezSingleCell and its manuscript to address the issues raised.

First, we agree that ezSingleCell does not implement any novel algorithm. Instead, the main objective of ezSingleCell and thus its key contribution is to make the analysis of single-cell and spatial omics data accessible to users without any programming background. This was our motivation in designing ezSingleCell's GUI and integration of analysis packages to achieve complete data analysis workflows. The majority of analysis packages are command line based and therefore require basic programming knowledge to efficiently employ and troubleshoot. Consequently, a well-designed GUI will make the software accessible to users without the necessary background. By integrating the various analysis software into a single package, we also reduce the effort required to select, install, and test the different software bundles and their dependencies. In ezSingleCell, we included software packages that are either the top performing algorithms in benchmark studies or highly cited popular algorithms. The other advantage of our software package integration is avoiding the effort needed to manage software packages and their dependencies. The matter of software package management is particularly notorious for Python based software, something that many users rely on environments like Conda to manage. Consequently, we believe that an integrated package can lower the hurdle for bench scientists without bioinformatics expertise to analyse their own data.

Moreover, we have revised ezSingleCell to allow **interplay between different modules**, allowing users to employ the output of a module in the data analysis of another module. For example, the output of cell types identified from single cell RNA-sequencing module can be used to (i) deconvolute the cell type proportions in spatial transcriptomics data, and (ii) perform cell type label transfer in single cell ATAC-seq data. To our best knowledge, no other existing web servers have this functionality. The implementation and workflow of module interactions are demonstrated in Figure 8 of the revised manuscript and Figure R1 in the response to comment #2) as follows.

2) This tool offers a range of modules that are independent of each other (e.g., 'single cell RNA-Seq', 'single-cell data integration', 'single cell ATAC-seq', etc.) to enable analysis and integration of different experimental data. Since the modules are independent, this software looks like several software glued together. For the analysis that each module would like to perform, there are nowadays a lot of packages that perform similar, or even more detailed functions. Therefore, for the current version, it is hard to see the indispensability of this software. However, the unique status of the software might be demonstrated if the authors could go further to implement modules and function interaction (like what they mentioned in the discussion).

Response: We thank the reviewer for the excellent suggestion on enabling interaction between the analysis modules. We have now added implemented module and function interaction in the revised version of ezSingleCell. Users can now employ the analysis output of certain modules as input in other modules, such as using the output of the cell types identified from single cell RNA-sequencing module to: (i) deconvolute the cell type proportions in spatial transcriptomics data, (ii) perform cell type label transfer in single cell ATAC-seq data, or transfer the phenotype label from the scIntegration module to (iii) the Spatial Transcriptomics or (iv) scATAC-seq module (Figure R1 below and Figure 8 of the revised manuscript). To the best of our knowledge, there are no web servers that have this utility to link two omics data types together in a unified interface.

ezSingleCell currently implements four types of interactions (Figure R1). In the scRNA-seq module at the step of cell type identification, users can click (1) the "Go to Spatial Deconvolution" button or (2) the "Annotate cell types for ATAC-data" button (Figure R2A). The first option will lead them to the deconvolution step of the Spatial Transcriptomics module. Here the annotated scRNA-seq data will be used as a reference to deconvolute the cell type proportions in spatial data using algorithms such as Seurat or GraphST. The second option will take the users to the cell type identification step of the scATAC-seq module. Here the Signac package is used to perform cell type label transfer from scRNA-seq to scATAC-seq. Moreover, we enabled two-way interactions. In the Spatial Transcriptomics module, at the deconvolution step, users can click on the "Load and process user reference dataset" option that will direct the users to the scRNA-seq module wherein users can upload, analyse, and annotate their reference scRNA-seq data (Figure R2B). The resulting annotated data can be used in

the Spatial Transcriptomics module for cell type deconvolution. Similarly, the “Load and process user reference dataset” button at the cell type identification step of scATAC-seq module allows users to go to the scRNA-seq module to analyse and annotate the input scRNA-seq (Figure R2C). The cell type labels are then transferred to scATAC-seq. Furthermore, we also enabled phenotype label transfer from the scIntegration module to the Spatial Transcriptomics and scATAC-seq modules.

As an example, we applied the cross-module interaction to deconvolute the cell types present in each spot by mapping cells from an scRNA-seq human breast tissue atlas onto the breast cancer 10x Visium dataset. We also performed phenotype label transfer by spatial mapping of cells from adjacent normal and solid tumour sites. As expected, we found that normal adjacent cells were mainly mapped onto the Healthy and Tumour Edge regions. The DCIS/LCIS regions showed higher levels of adjacent normal cell mapping than IDC regions. Conversely, the solid tumour-derived cells mostly mapped to the IDC regions.

Figure R1. Cross module interactions in ezSingleCell.

Figure R2. Graphical user interface for cross module interactions.

3) It is difficult to conclude whether this all-in-one software can really help the researchers' work and save their time. First of all, the software only called some of the functions of some popular single-cell analysis packages and fixed most of the adjustable parameters, meaning many functions carried by the original package are not able to use (for instance, it is not possible to specify only two cell groups for DGE analysis). Secondly, this integrated pipeline leads to the need to restart the program from the first step again if something goes wrong at an intermediate step. Thirdly, it is also impossible to modify or delete any clustering or prediction results, even if we found some of them, even if we found that some inferences may be not reasonable biologically.

Response: We thank the reviewer for the point on available functionality and customizability of functions in ezSingleCell. Our focus with ezSingleCell is to ensure important and commonly used functions to be available for the user. However, we appreciate the point on giving users the flexibility and breadth of functionality of the underlying packages. Therefore, we have worked on increasing available options to enhance flexibility. For example, users can subset for a particular cluster of interest and save it in an RDS file that can be further analysed by ezSingleCell. We have also incorporated a few more functions in ezSingleCell such as cell cycle scoring analysis, and pairwise DEG analysis where users can specify only two cell groups or clusters for DGE analysis. We are also working on enabling users to select a group of cells by drawing a rectangle or using a lasso tool function.

Our selection of input parameters for functions is also similarly motivated to ensure ease of use with commonly customized options. We will add the remaining parameters in expandable menus to enable full customization for interested users. We have also improved the interface so that users can now navigate to previous steps of analysis to change certain parameters of interest to re-execute the analysis. This is very useful from the biological exploration point of view. We have also updated the clustering functionalities for greater flexibility. Users can sub-cluster a cluster of interest according to user defined parameters (such as clustering resolution) which would aid in obtaining subsets of cell type populations. A snapshot of sub-cluster analysis is shown below (Figure R3).

Figure R3. Sub-cluster a cluster of your interest according to user-defined parameters

In the above example, we selected the cluster of Memory CD4 T cells identified by CELLiD and sub-clustered them at a resolution of 0.6 to get 4 sub-clusters (Memory CD4 T cell_0, Memory CD4 T cell_1, Memory CD4 T cell_2, Memory CD4 T cell_3).

User can now rename clusters, thus giving the option to eliminate cluster that the user deems as incorrect. Users can perform cell type annotation using CELLiD and thereafter modify the cluster labels in the cell type prediction table. Finally, users can click on 'Rename clusters' to rename the cluster of interest. A stepwise demonstration is shown as below in Figure R4:

A. Users can modify the cluster labels in the cell type prediction table and rename clusters.

Original ID	newID
Naive CD4 T cell	Naive CD4 T cell
Memory CD4 T cell	Memory CD4 T cell
Naive B cell	Naive B cell
GZMB CD8 T cell	GZMB CD8 T cell
CD14 monocyte	CD14 monocyte
CD14 monocyte	CD14 monocyte
GZMK CD8 T cell	GZMK CD8 T cell
CD16 NK cell	CD16 NK cell
CD16 monocyte	CD16 monocyte
Dendritic cell	Dendritic cell
Megakaryocyte	Megakaryocyte

B. Click on the cell in the table to rename.

Original ID	newID
Naive CD4 T cell	Naive CD4 T cell
Memory CD4 T cell	Memory CD4 T cell
Naive B cell	Naive B cell
GZMB CD8 T cell	GZMB CD8 T cell
CD14 monocyte	CD14 monocyte
CD14 monocyte	CD14 monocyte
GZMK CD8 T cell	GZMK CD8 T cell
CD16 NK cell	CD16 NK cell
CD16 monocyte	CD16 monocyte
Dendritic cell	Dendritic cell
Megakaryocyte	Megakaryocyte

C. Rename the cell in the table according to your choice.

D. The clusters will be renamed with the new label.

Figure R4. A stepwise demonstration of modifying clustering and cell type prediction results.

We have now fixed the ezSingleCell webserver so that users do not need to restart the program if something goes wrong at an intermediate step.

Finally, we appreciate reviewer's suggestions, but it would not be feasible to add too many utilities right now as our aim is to aid the bench scientists with no programming experience in performing basic and advanced analysis on a standard desktop computer. We aim to add more features in the near future based on biological importance and popularity of use.

4) With rapid advances in biotechnology and computational methods, there are dozens of algorithms that perform the same or similar functions. However, the authors did not declare why they chose to call the package specified in the article. For example, why use CELLiD to identify cell types, and why use CellphoneDB to compute cell interactions? Is this due to the considerations of time complexity, space complexity, or accuracy?

Response: We agree with the reviewer that new methods are being constantly published and that there are many methods that do serve the same or similar functionalities. For the tasks available within ezSingleCell, we carefully selected the best performing and reliable methods according to benchmarking papers from literature as well as their usage popularity or citation.

For the task of batch integration, our benchmarking of batch effect correction methods indicated that Harmony, LIGER, and Harmony are the top 3 performers [1]. In a more recent benchmarking study, Luecken et al. recommended scANVI, Scanorama, scVI, fastMNN and scGen for large scale data integration [2]. Among these methods, Seurat, Harmony and scVI have gradually gained popularity and been widely adopted by researchers. As such, we selected Seurat, Harmony, and scVI to be included in ezSingleCell. Among these three, we believe most users should be able to find one that best suits their data and analysis needs.

For the task of cell type identification, we included CELLiD and CellTypist2. CELLiD is an atlas guided automatic cell type identification tool developed in house as part of the DISCO platform [3]. We believe that for successful automatic cell type annotation, a high-quality cell type reference database is necessary, and have built such reference by integrating more than 51 million cells from 11968 samples, covering 107 tissues/cell lines/organoids, 158 diseases, and 20 platforms. CellTypist2 is also included as it is currently the most popular tool for cell type annotation [4].

For cell type deconvolution of spatial transcriptomics, ezSingleCell currently employs our in-house deep learning method, GraphST, that we have benchmarked and found to be competitive with other leading published methods. We are also working to include cell2location and RCTD, which have been highlighted by two benchmarking studies [5,6].

For single-cell multi-omics analysis, we chose Seurat and MOFA+ based on in-house unpublished benchmarking. We also want to highlight that ezSingleCell is not static; it will be continually updated and refined, including the integrated packages. As newer and better methods emerge, we also plan to include them after careful testing and benchmarking. Finally, the user has the option in ezSingleCell to download the data object used in the analysis. Advanced users can employ other tools for offline analysis, as well as engage other experts/bioinformaticians for such analyses.

1. Tran, H.T.N., Ang, K.S., Chevrier, M. et al. A benchmark of batch-effect correction methods for single-cell RNA sequencing data. *Genome Biol.* 21, 12 (2020) recommends the use of **Harmony, LIGER, and Seurat 3 for data integration.**
2. Luecken, M.D., Büttner, M., Chaichoompu, K. et al. Benchmarking atlas-level data integration in single-cell genomics. *Nat Methods* 19, 41–50 (2022) recommends the use of **scANVI, Scanorama, scVI, fastMNN and scGen for large-scale data integration.**
3. Li, M., Zhang, X., K.S., Ang, et al. DISCO: a database of Deeply Integrated human Single-Cell Omics data. *NAR* 50, D596–D602 (2022), CELLiD is an in-house cell type identification algorithm developed by our lab for prediction using a reference dataset prepared by integrating more than 51 million cells from 11968 samples, covering 107 tissues/cell lines/organoids, 158 diseases, and 20 platforms.
4. Domínguez Conde, C., Xu, C., Jarvis, L.B., et al. Cross-tissue immune cell analysis reveals tissue-specific features in humans, 376, eabl5197 (2022), established a high quality cell type reference to be used in conjunction with a machine learning tool for cell type annotation. Current version is version 2.
5. Li B, Zhang W, Guo C, Xu H, Li L, Fang M, Hu Y, Zhang X, Yao X, Tang M, Liu K, Zhao X, Lin J, Cheng L, Chen F, Xue T, Qu K. Benchmarking spatial and single-cell transcriptomics integration methods for transcript distribution prediction and cell type deconvolution. *Nat Methods.* 2022 Jun;19(6):662-670 recommends the use of **cell2location, SpatialDWLS and RCTD.**
6. Li H, Zhou J, Li Z, Chen S, Liao X, Zhang B, Zhang R, Wang Y, Sun S, Gao X. A comprehensive benchmarking with practical guidelines for cellular deconvolution of spatial transcriptomics. *Nat Commun.* 2023 Mar 21;14(1):1548 recommends the use of CARD, **cell2location, Tangram and RCTD.**

5) How could users evaluate the different clustering results obtained from different parameters in cell clustering analysis, and determine the optimal number of clusters?

Response: We agree that the selection of clustering resolution can be a difficult decision to make. It is indeed a common issue which is dependent on the characteristics of data and end goal of the analysis. Therefore, we have now incorporated clustree (<https://github.com/lazappi/clustree>) to assist users in selecting a suitable number of clusters (Figure R5). We also provide a detailed tutorial to guide the users to plot marker expression of known cell types to aid in selecting a suitable clustering resolution.

Figure R5. Selection of clustering resolution in ezSingleCell.

In general, I encourage the authors to reflect on the innovation of this work and to put themselves in the users' shoes to design the software's architecture. Unfortunately, in its current state, I do not recommend accepting this manuscript.

Thank you for this valuable suggestion. We appreciate your efforts to carefully go through our ezSingleCell webserver and suggest points for improvement.

Reviewer #2 (Remarks to the Author):

Sethi et al., presented an interactive tool for single-cell and spatial omics analysis, namely ezSingleCell. By assembling current state of art bioinformatic methods, it aims to enable bench scientist for hassle-free in-depth analysis. Compared to other tools with similar purposes, ezSingleCell offers more functionalities including cell-cell communication, spatial and scATAC-seq analysis. And this is facilitated by a web server and/or shinyApp software that runs on a PC. I find the tool may help bench scientists to run their single-cell and spatial data and have some initial explorations or even publication ready figures. But the current version appears not mature yet, and further effort and resources are required.

Firstly, we would like to thank the reviewer for the helpful suggestions on improvements and bugs spotted. We are also happy that the reviewer appreciates the usefulness of ezSingleCell to bench scientists in analysing their data. To improve ezSingleCell, we have revised the functionalities of ezSingleCell to increase its functionality to users as listed below:

1. We have revised ezSingleCell to allow interplay between different modules, where users can employ the output of a module in the data analysis of another module. For example, the output of cell types identified in the single cell RNA-sequencing module can be used to (i) deconvolute the cell type proportions in spatial transcriptomics data, and (ii) perform cell type label transfer in single cell ATAC-seq data. To our best knowledge, no other existing webserver have this functionality.

2. We have expanded ezSingleCell to incorporate more methods such as scVI for single-cell RNA-seq data integration and CellTypist for automated cell type annotation. Currently we are working on incorporating totalVI for multi-omics analysis, and RCTD and cell2location for spatial transcriptomics analysis.

3. We have fixed various bugs that result in errors and prevent the smooth running of the data processing pipelines.

4. We have incorporated clustree (<https://github.com/lazappi/clustree>) to assist users in selecting a suitable number of cell clusters. Users can also merge clusters at the cell type annotation stage by renaming them as appropriately (Figure R4). In the future, users will be able to select a cluster and subset it according to their own choice of parameters.

5. We have added functionality for filtering cells at the QC stage for all 5 modules. For example, users can filter cells by mitochondrial percentage or minimum number of features, as well as filter features by a minimum number of cells.

6. We have also significantly increased the available server resources to better handle the workload required as listed below:

- a. We have moved ezSingleCell to a new hosting server with more compute resources to facilitate analysis of larger datasets by more users in parallel. We have upgraded the server with 2 GPUs (GeForce RTX 3060) to accelerate some analyses, especially those that employ deep learning such as GraphST and scVI. Each GPU offers 12 GB GDDR6 graphics memory with a 192 Bit memory bus.
- b. We have enabled large data analysis by incorporating the sketch package. Geometric sketching is a method to subsample massive scRNA-seq datasets while preserving rare cell states. "Sketching" is useful for accelerating clustering, visualization, and integration analyses. Analysis of 50,000 cells can be done in 4-5 minutes while analysis of 100,000 cells can be done in 10 minutes.

Major points:

1) *It runs well when "Load example and run", but did not work when I uploaded local single-cell data, for example a h5 file as stated in the manuscript, and it raised error "Error: argument is of length zero" (google chrome browser)*

Response: Thank you for pointing out this error. It is a bug which has been fixed now.

2) *The server appears not stable as even loading of example data, running cellid function already slowdowned or crashed the server. Though it is understandable that such analysis may require high computational resource. With more than 3000 cells (often 100k cells for almost ready study cases nowadays) and concurrent users, and I think it can hardly work without massive infrastructures. I suggest the authors to specify what they can offer for this server, and what is their limit.*

Response: Thank you for pointing out this issue. We have now migrated our webserver to a new server which has significantly stronger infrastructure capabilities for running jobs from 3 users at the same time. The new server is also equipped with 2 GPUs (GeForce RTX 3060) that can easily speed up computing-intensive algorithms to achieve shorter runtimes.

We have also incorporated the sketch package to subsample massive scRNA-seq datasets while preserving rare cell states. "Sketching" is useful for accelerating clustering, visualization, and integration analyses. Analysis of 50,000 cells can be done in 4-5 minutes while analysis of 100,000 cells can be done in 10 minutes.

We also provide ezSingleCell as a software package with a shinyApp interface (<https://github.com/JinmiaoChenLab/ezSingleCell2>) that allows users to run it on their own computers when the ezSingleCell server is busy.

To speed up CELLiD, the user needs to choose a relevant tissue reference atlas, which will significantly reduce the runtime. By default, all DISCO reference atlases are used and that slows down the process. We will add an instruction note to guide the users in running CELLiD.

3) *In the single-cell data integration “Quality control” section, no threshold setting option to do the filtering, so actually no quality control. This apply to some other functionalities, where few option is given.*

Response: Thank you for pointing out this issue. We have modified the interface to include thresholding options to filter out low quality cells for all 5 modules in ezSingleCell.

4) *Line 166 and supplementary Figure 3F: KLRB1 is not a good MAIT marker, instead, the author should use SLC4A10 etc. The MAIT annotation seems not correct not only because of the wrong marker used, but also the proportion was way higher than expected if the authors’ annotation is used. In another word, the accuracy of the pipeline is questioned.*

Response: Thank you for this suggestion and we agree with the reviewer that MAIT cell abundance should be much lower. We apologize for this mistake from our cell type identification algorithm. Our in-house algorithm (CELLiD) performs cluster level cell type annotation wherein it calculates the average expression for each cluster and thereafter performs correlation analysis with reference data from DISCO’s cell type database (<https://www.immunesinglecell.org/>). As such, the cell type prediction result is highly dependent on the clustering resolution. The FindClusters() function implemented in Seurat contains a resolution parameter that sets the ‘granularity’ of the downstream clustering, with increased values leading to a greater number of clusters. In our manuscript, we set the clustering resolution to 2 and obtained 20 clusters. Due to the high clustering resolution, the GZMB CD8 T cell population was split into 2 clusters that were predicted to be GZMB CD8 T cell and MAIT cells respectively. By checking the expression of SLC4A10, we did observe some MAIT cells in that cluster. We have now decreased the cluster resolution from 2 to 1.5 and observed the MAIT cell cluster is merged to GZMB CD8 T cell (Figure R6). We have also updated the figures in the revised manuscript accordingly.

Here we would like to mention that ezSingleCell is a flexible and user-oriented software where users can alter the clustering resolution and rename the cell type annotation according to their domain knowledge and biological question. We have also realized that in most cases, it is necessary to perform manual annotation or at least manual curation of the automated annotation based on marker gene expression. In ezSingleCell, we provide the users the facility to rename clusters manually after performing manual annotation by either looking at DEGs or checking known marker genes. To the best of our knowledge, there is no one perfect method that can give very accurate and reliable annotation for all datasets. As such, we have included CellTypist2 to enable cell-level annotation and as an alternative. With automated tools like CELLiD and CellTypist, bench scientists will obtain a draft annotation and then refine it easily using ezSingleCell’s GUI.

Figure R6. Clustering and cell type annotation at a lower resolution.

Minor points:

1) Line 109: table 4 appears before table 2 and 3

Response: Thank you for spotting this error. We have corrected this.

2) For the interface design, I noticed some pages showed multiple buttons, since its end users are researchers without any programming experience, I suggest the authors to put numbers on the buttons to have a clear order, which prevents ambiguous mistakes.

Response: Thank you for pointing out this issue. We have now added numbers in the side panel box for each module (as shown in Figure R7).

Figure R7. New GUI with numbers added to the buttons in the side panel box.

3) Line 167: Naive and memory B cell markers need to be specified in the figure. In addition, it seems not difficult to split naïve and memory given the cell amount used in this particular dataset.

Response: Thank you for pointing out this issue, and we agree with the reviewer that it is not difficult to split naïve and memory B. We used a higher clustering resolution to obtain finer clusters and split the B cells into naïve and memory B cells. We have now added Naïve and memory B cell markers in supplementary Figure 3F, 3G.

4) Line 229: figure 7F appears before 7E, such order mistakes appear several times.

Response: Thank you for spotting errors within the manuscript. We have revised the figures to ensure that they are presented in the correct order.

5) Layout of some Figures should be carefully aligned or names. i.e. Figure 3 and 7F, as well as the legends, with a bit more information.

Response: Thank you for highlighting this issue. We have carefully adjusted, aligned, and renamed the figures to ensure that they are well organized and clear to the readers.

Reviewer #3 (Remarks to the Author):

Sethi et al. developed ezSingleCell, a convenient and easy web service for single cell and spatial analysis without requiring prior programming knowledge. Overall, this study could benefit researchers without bioinformatics expertise or with limited access to computing resources. Meanwhile, it could probably aid the study of single-cell and spatial omics. In general, I would like to make the following suggestions for improvement.

We are grateful for the useful comments from the reviewer to improve ezSingleCell. We are also happy that the reviewer appreciates the usefulness of ezSingleCell to bench scientists in analysing their data. We have moved ezSingleCell to a more powerful hosting server to ensure the sufficient of compute resources to users, as well as its continued availability to users. The new hosting server has more compute resources to facilitate analysis of larger datasets by more users in parallel. The server is also equipped with 2 GPUs (GeForce RTX 3060) to accelerate some analyses especially those employ deep learning such as GraphST and scVI. Each GPU in our server offers 12 GB GDDR6 graphics memory with a 192 Bit memory bus. As with our earlier work, DISCO: a database of Deeply Integrated human Single-Cell Omics data, we will continue to improve and extend ezSingleCell to serve as a comprehensive web service for analysing single-cell and spatial data.

As per suggestion, we have added a more comprehensive benchmark with larger cell counts of 50,000 or 100,000 cells, and pathway enrichment tools like fgsea. We also designed ezSingleCell to be as species agnostic as possible, with species restriction applicable to the cell type annotation and pathway analysis functionalities since they are dependent on their underlying reference data.

In addition, we have added various functionalities as suggested by other reviewers. Users can use the output of some modules for others, such as the processed output of the single-cell RNA-seq module as input to the spatial transcriptomics module for annotation. We have also added clustree (<https://github.com/lazappi/clustree>) to help guide the selection of single-cell clustering resolution and allow users to merge clusters to modify the overall clustering. Finally, we have added QC filters in all 5 modules of ezSingleCell for cell quality filtering and increased the availability of parameters for the different functions for greater customization by users.

1)The authors should specify their plan to maintain and upgrade ezSingleCell. Quite a few websites are no longer timely and efficiently upgraded after publication. How can authors guarantee the efficient maintenance of ezSingleCell in five years.

Response: We understand the concern on ezSingleCell's availability and maintenance over time, which is important for users to ensure the continuity and integrity of their analysis. For illustration, we have previously introduced DISCO, a web resource for processed single-cell data and constructed single-cell atlases (website: <http://www.immunesinglecell.org/>, manuscript: <https://academic.oup.com/nar/article/50/D1/D596/6430491>). DISCO was first established in 2021 and is being constantly updated in terms of newly published data, newly constructed atlases, as well as updating existing atlases and implementing new functionalities. We are committed to the same trajectory as DISCO with ezSingleCell, aiming to make ezSingleCell an established platform for

analysing single-cell and spatial data. Currently, ezSingleCell is hosted on the server provided by Vishuo Biomedical Pte. Ltd, with whom we have agreement for continued hosting.

Furthermore, there is also an offline version of ezSingleCell which is fully compatible with the data objects created during analysis in ezSingleCell and is downloadable by the user. This allows the user to engage in off-line data analysis or even perform other kinds of analysis with other tools.

2) Currently, a lot of online analytic tools (CancerSEA, SPEED, SPatialDB, Galaxy Single Cell Omics Workbench) have been developed. Authors are suggested to compare ezSingleCell with those already published tools, either in introduction or in discussion section, to highlight the uniqueness and novelty of ezSingleCell.

Response: We thank the reviewer for highlighting other works of online resources and tools relevant to single-cell and spatial transcriptomic data and data analysis. We have carefully checked the listed tools of CancerSEA (<http://biocc.hrbmu.edu.cn/CancerSEA/> or <http://202.97.205.69/CancerSEA/>), SPEED (<http://8.142.154.29> or <http://speedatlas.net>), SPatialDB (<http://www.spatialomics.org/SpatialDB/>), and Galaxy Single Cell Omics Workbench (<https://singlecell.usegalaxy.eu/>).

CancerSEA is purportedly an online database that describes distinct functional states of cancer cells derived from single-cell data. It hosts single-cell datasets and derived knowledge from the datasets, with search and visualization facilities for the hosted data. There appears to be no functionality to upload user data for analysis.

SPEED is an online database of single-cell pan-species atlas. It hosts single-cell data from 127 species, as well as single-cell atlases of evolution, development, and diseases. SPEED offers a data upload facility, but it appears to be a data submission feature, rather than uploading data for analysis. Consequently, we conclude that the analysis tools offered on SPEED are only applicable on the hosted data.

SPatialDB is an online database of spatial data, hosting data generated using a variety of techniques including Slide-seq, LCM-seq, seqFISH, and MERFish. It allows users to select hosted data for visual comparison of slides and gene expression comparison. There is a data upload facility, but it also appears to be a data *submission* feature, rather than uploading data for analysis.

The Galaxy Single Cell Omics Workbench is a comprehensive sequencing data analysis website that enables workflows starting from FASTA/FASTQ files. Users can perform genomic analysis including FASTQ QC, assembly, annotation, alignment, variant calling, etc. It also offers Scanpy functions for processing user uploaded single-cell RNA-seq data.

Since ezSingleCell is an online platform designed to enable online analysis of user uploaded data with a GUI, we therefore do not think that CancerSEA, SPEED, and SPatialDB are suitable tools for comparison. We believe only Galaxy Single Cell Omics Workbench is relevant for comparison. A particular deployment of the Galaxy framework is SCiAp by Moreno et al. that offers access to HCA and SCEA's data, as well as user uploaded data for analysis. As the single-cell analysis functionalities offered are essentially identical between the two Galaxy instances and SCiAp is already listed in Table 1 for comparison, additional comparison will be redundant.

3) Authors should test the performance of ezSingleCell on larger data sets (cell number between 50,000 and 100,000).

Response: As suggested, we have expanded our performance benchmarking to cover a larger range of cell counts. Here, we tested ezSingleCell on a breast cancer dataset downloaded from the DISCO website (<https://immunesinglecell.org/>) and tested ezSingleCell on a range of cell counts (as shown in Table R1 below).

S. No	# of cells/cluster	# of cells	Time (in min)
1	5	195	1
2	10	390	1
3	20	780	1
4	30	1170	1
5	50	1914	1.5
6	75	3714	1.5
7	100	5508	2
8	200	7258	2
9	300	10595	2.2
10	500	16608	2.5
11	750	23525	3.5
12	1000	29887	4
13	1500	41600	5
14	2000	52159	6
15	3000	67728	8
16	4000	79957	10
17	5000	91876	14
18	6000	102876	16

Table R1. Table demonstrating the Run-time performance of ezSingleCell at different cell counts

The results shows that ezSingleCell can handle dataset of 100,000 cells, taking around 16 mins (Figure R8) to perform basic single cell RNA-seq analysis comprising of variable gene selection, clustering, dimension reduction, and cell type identification using CELLiD. However, for large datasets the data uploading may vary and is expected to be slow for larger dataset depending on network capacity.

Figure R8. Benchmarking run-time of ezSingleCell at different cell counts

4) I am curious about the performance of ezSingleCell on other scRNAseq techniques, for example sci-RNA-seq3.

Response: ezSingleCell offers the same support for other techniques. We have newly updated ezSingleCell to handle sci-RNA-seq3 data. In the below example, we tested ezSingleCell on a sci-RNA-seq3 thymus dataset generated by Cao J *et al*, 2020, published in *Science* (<https://descartes.brotmanbaty.org/>) (as shown in Figure R9 and Figure R10). This illustrates that ezSingleCell can handle data acquired with sci-RNA-seq3.

Human Gene Expression During Development

4M Cells	121 Tissues	15 Organs
----------	-------------	-----------

Human Chromatin Accessibility During Development

720K Cells	53 Tissues	15 Organs
------------	------------	-----------

Mammalian Organogenesis

~2M Cells	61 Embryos
-----------	------------

Download

Raw Gene Count Sparse Matrices
Sparse gene (row) by cell (column) matrices split by organs.

ADRENAL CELLS (69M RDS)	CEREBELLUM CELLS (27C RDS)	CEREBRUM CELLS (36G RDS)	EYE CELLS (70M RDS)
HEART CELLS (152M RDS)	INTESTINE CELLS (83M RDS)	KIDNEY CELLS (360M RDS)	LIVER CELLS (148M RDS)
LUNG CELLS (328M RDS)	MUSCLE CELLS (43M RDS)	PANCREAS CELLS (98M RDS)	PLACENTA CELLS (32M RDS)
SPIKE IN 293T/3T3 CELLS (63M RDS)	SPLEEN CELLS (24M RDS)	STOMACH CELLS (16M RDS)	THYMUS CELLS (11M RDS)

Sampled Data
To compare cell types across organs, we randomly sampled 5,000 cells per cell type per organ (or in cases where less than 5,000 cells of a given cell type were represented in a given organ, all cells were taken). The data comprise gene count information including a total of 377,456 cells.

Figure R9. Example of sci-RNA-seq3 data.

Figure R10. Analysis of sci-RNA-seq3 with ezSingleCell.

5) Could the author integrate target prediction and GO/KEGG enrichment function into the scATAC-seq module of ezSingleCell?

Response: This suggestion to include target prediction and pathway enrichment functionalities is good and relevant to ezSingleCell. We have added the feature to link peaks to genes and visualize the signals in each cluster for a specific gene of interest (as shown in Figure R11). We have also added GO/KEGG enrichment analysis for single cell ATAC-seq data (as shown in Figure R12).

Figure R11. Target prediction function of the single-cell ATAC-seq module.

Figure R12. GO/KEGG enrichment function of the single-cell ATAC-seq module.

6) How many species does ezSingleCell support? Could ezSingleCell be employed to analyze the data of other species (e.g., fruit fly, zebrafish, worm)? Have the authors tested what problems it might happen when dealing with scRNAseq data from other species. Do the authors have any plan to solve those problems?

Response: ezSingleCell has been designed to be as species agnostic as possible. As such, only the cell type annotation and pathway analysis functionalities are species restricted as they both rely on underlying knowledge resources to function. Currently, the cell type annotation function CELLiD supports human and mouse, and users can upload their own cell type transcriptome to serve as reference. For CellTypist, the package currently only contains human cell reference and therefore only supports human cell type annotation. Nevertheless, users are also allowed to manually annotate the cell types by plotting known marker genes and visualizing them. For the GSEA analysis functionality, it is also reliant on the reference gene sets available. As the reference uses human gene symbols, the user uploaded data has to be annotated with human gene symbols as well. For the newly added pathway analysis, the supported reference is also tied to the respective reference database available in the original software packages.

Nevertheless, the suggestion to increase species coverage is a good suggestion and we will incorporate cell type references for other popular model organisms such as the fruit fly, zebrafish, and worm in the next version of ezSingleCell.

Reviewer #1 (Remarks to the Author):

Sethi et al. integrated several multi-omics, spatial, and single-cell analysis tools to develop ezSingleCell software. The authors claim that the primary contribution of this work is to enable users without any programming background to analyze single-cell and spatial omics data. In response to reviewer comments, the revised manuscript attempts to address a number of issues. For example, ezSingleCell 1) adds the functionality of some inter-module interactions, 2) integrates more full microanalysis methods, and 3) completes a "cluster tree" to support the user in finalizing the optimal number of clusters. However, there are still some issues that have not yet been resolved.

Major points

1) The main issue lies in the confusing overall architecture of the software. For instance, the "Single Cell Data Integration" module has been modified to have very detailed functionality, while other modules have very limited capabilities of downstream analyses. Multiple downstream analyses (e.g. UMAP, clustering, sub-clustering, DEGs, GSEA, CCC) are actually completely common across modules. The authors could consider sharing downstream analysis functionality across all modules to enable crosstalk between modules.

2) The confusing architecture of the software is also reflected in the organization of the steps in modules. The authors have numerically arranged the functions. However, it is challenging for users to discern which functions are optional, parallel, or require sequential execution. In addition, some steps (or functions) could be consolidated to make the entire process more comprehensible, such as merging DEGs with DEG pairs and clustering with sub-clustering.

Minor Point:

1) The names of cell types overlapped on the UMAP plot

2) It will be better if users are able to search for the genes they want to visualize on their own. The current gene list is too long to find a specific gene.

3) Users should not be allowed to change the names of the original cell types (in the Single Cell Data Integration module).

Reviewer #2 (Remarks to the Author):

The authors have taken care to address all my previous concerns in detail. And I am satisfied with most of the improvement. Below are some minor points to be further addressed before going forward.

1.It seems that the Fig. S3B is not updated since the MAIT population is not removed.

2.By checking the source code website <https://github.com/JinmiaoChenLab/ezSingleCell2>, I notice the last update was from 7 months ago, I suppose it was still the old version, thus the code improvement can not be judged.

3.In some step such as "8. Deconvolution", it shows "Error:...." already before any action. Similar situation happens also in other modules.

4.Single-cell ATAC-seq pipeline step "10.Gene Set Enrichment Analysis" takes either very long or buggy, please make sure it is still functioning.

Reviewer #3 (Remarks to the Author):

All my concerns have been addressed

Response to comments for paper NCOMMS-23-12483-T “ezSingleCell: An integrated one-stop single-cell and spatial omics analysis platform for bench scientists”

Thank you to all reviewers for your constructive suggestions and feedback. We are very grateful for your time spent in reviewing the manuscript and testing the ezSingleCell software. Your comments have helped us to greatly improve ezSingleCell. We have extensively updated ezSingleCell based on your feedback. The changes implemented are as follows:

1. Implemented detailed functionality such as DEG analysis, cell type similarity, Gene Set Enrichment Analysis (GSEA), cell-cell communication for other modules in addition to the single cell RNA-seq module.
2. We have added a feature that makes it easier for users to discern which functions are optional, parallel, or sequential.
3. We have merged the steps that perform the same or similar functionality. For example, in the single cell RNA-seq module, we have merged: (i) cell type specific DEG analysis and pairwise DEG analysis; (ii) clustering and sub-clustering analysis.
4. We have added methods such as rGREAT for Gene Set Enrichment Analysis for scATAC-seq data. rGREAT infers biological functions directly from peaks as opposed to ‘fgsea’ that uses gene activity to infer biological functions.
5. We have fixed minor bugs such as long run time in Gene Set Enrichment Analysis for scATAC-seq data, displaying an error message on the screen when user doesn’t perform any action.
6. We have updated the downloadable ezSingleCell package at <https://github.com/JinmiaoChenLab/ezSingleCell2>.

Comments from the Reviewer

Reviewer #1 (Remarks to the Author):

Sethi et al. integrated several multi-omics, spatial, and single-cell analysis tools to develop ezSingleCell software. The authors claim that the primary contribution of this work is to enable users without any programming background to analyze single-cell and spatial omics data. In response to reviewer comments, the revised manuscript attempts to address a number of issues. For example, ezSingleCell 1) adds the functionality of some inter-module interactions, 2) integrates more full microanalysis methods, and 3) completes a "cluster tree" to support the user in finalizing the optimal number of clusters. However, there are still some issues that have not yet been resolved.

Major points

1) The main issue lies in the confusing overall architecture of the software. For instance, the "Single Cell Data Integration" module has been modified to have very detailed functionality, while other modules have very limited capabilities of downstream analyses. Multiple downstream analyses (e.g. UMAP, clustering, sub-clustering, DEGs, GSEA, CCC) are actually completely common across modules. The authors could consider sharing downstream analysis functionality across all modules to enable crosstalk between modules.

Response: Thank you very much for the valuable inputs. We agree that multiple downstream analyses are common across modules. We have now modified ezSingleCell to have detailed functionality such as DEG Analysis, CellType Similarity, GSEA, Cell-Cell Communication for other modules in addition to the single cell RNA-seq module. Now the different modules share the same downstream analysis functionality, except for some functions that are module specific, such as Peak2GeneLinkage for the scATAC-seq module and Deconvolution for the spatial module.

2) The confusing architecture of the software is also reflected in the organization of the steps in modules. The authors have numerically arranged the functions. However, it is challenging for users to discern which functions are optional, parallel, or require sequential execution. In addition, some steps (or functions) could be consolidated to make the entire process more comprehensible, such as merging DEGs with DEG pairs and clustering with sub-clustering.

Response: Thank you very much for this important comment. We agree with it and have now modified ezSingleCell to make it easier for users to discern which functions are optional, parallel, or sequential. Specifically, we added a new feature to enable one tab only (i.e. Upload your data) and disable rest of the tabs (shown below) at the beginning.

When the current step is completed, we enable the next tab while disabling rest of the tabs. Likewise, we enable the tabs based on the completion of the previous step in the order of sequential execution (shown below).

For steps that do not require sequential execution, we enable multiple tabs based on the completion of the previous step. For example, when performing scRNA-seq analysis, once we complete PCA analysis, we can perform UMAP, tSNE, or Clustering in any order. Hence, we enable all these tabs after the completion of PCA (shown below).

As suggested by the reviewer, we have also merged the steps that perform similar tasks such as merging cell type specific DEG analysis with pairwise DEGs, and clustering with sub-clustering (as shown below).

Merging CellType specific DEGs and Pairwise DEGs -

Single cell RNA-Sequencing | Single cell data integration | Spatial Transcriptomics | Single cell multiomics | Single cell ATAC-seq | Help

Overview

1. Upload your data
2. Normalization and Variable Feature Selection
3. PCA
4. UMAP
5. tSNE
6. Clustering
7. Cell type identification
8. Celltype similarity
9. DEGs

Type of DEG analysis

Celltype specific

min.pct: 0.25 | logfc.threshold: 0.25 | Test use: wilcox

Run DEGs

Single cell RNA-Sequencing | Single cell data integration | Spatial Transcriptomics | Single cell multiomics | Single cell ATAC-seq | Help

Overview

1. Upload your data
2. Normalization and Variable Feature Selection
3. PCA
4. UMAP
5. tSNE
6. Clustering
7. Cell type identification
8. Celltype similarity
9. DEGs

Type of DEG analysis

Celltype specific

Group by: primary.predict | min.pct: 0.5 | logfc.threshold: 0.5 | Test use: wilcox

Run DEGs

Show 10 entries | Search:

	p_val	avg_log2FC	pct.1	pct.2	p_val_adj	cluster	gene
CCL5	1.790543351161191e-169	2.874756117842729	0.966	0.249	2.455551151782457e-165	GZMB CD8 T cell	CCL5
NKG7	5.639208567827553e-162	2.212145299417807	0.97	0.23	7.733610629918706e-158	GZMB CD8 T cell	NKG7
CST7	4.651292619906309e-158	2.026025399927761	0.817	0.131	6.378782698939512e-154	GZMB CD8 T cell	CST7
GZMK	6.066416827168678e-155	2.782349120291872	0.605	0.066	8.319484036779125e-151	GZMB CD8 T cell	GZMK
GZMA	4.085931646317311e-149	1.979220971371666	0.814	0.134	5.60344666250236e-145	GZMB CD8 T cell	GZMA
CD8A	1.348952117299274e-105	1.883945891822834	0.525	0.075	1.849952933664224e-101	GZMB CD8 T cell	CD8A

Pairwise DEGs:

Single cell RNA-Sequencing | Single cell data integration | Spatial Transcriptomics | Single cell multiomics | Single cell ATAC-seq | Help

Overview

1. Upload your data
2. Normalization and Variable Feature Selection
3. PCA
4. UMAP
5. tSNE
6. Clustering
7. Cell type identification
8. Celltype similarity
9. DEGs
10. Data visualization
11. GSEA
12. Cell-cell communication

Type of DEG analysis

Pairwise DEGs

Group by: primary.predict | Celltype1: CD16 monocyte | Celltype2: CD14 monocyte | Run Pairwise DEGs

min.pct: 0.25 | logfc.threshold: 0.25 | Test use: wilcox

Volcano plot
Enhanced Volcano

Legend: NS (grey), Log2 FC (green), p-value (blue), p-value and log2 FC (red)

total = 670 variables

Merging Clustering and Sub-Clustering –

Clustering:

Sub-Clustering:

Minor Point:

1) The names of cell types overlapped on the UMAP plot

Response: Thank you for pointing this out. We have fixed it and now the cell type names do not overlap on the UMAP plot.

2) It will be better if users are able to search for the genes they want to visualize on their own. The current gene list is too long to find a specific gene.

Response: Thank you for this suggestion. Yes, users can type the gene name and search for their gene of interest as shown below.

3) Users should not be allowed to change the names of the original cell types (in the Single Cell Data Integration module).

Response: Yes, we keep both the original cell type names and the cell type predicted by the algorithms CELLiD and CellTypist. The original cell types are stored as 'Metadata Celltype' whereas the cell type predicted by CELLiD and CellTypist are stored as 'Predicted Celltype'.

Reviewer #2 (Remarks to the Author):

The authors have taken care to address all my previous concerns in detail. And I am satisfied with most of the improvement. Below are some minor points to be further addressed before going forward.

1.It seems that the Fig. S3B is not updated since the MAIT population is not removed.

Response: Thank you for alerting us to this mistake. We have updated Fig. S3B accordingly.

2.By checking the source code website <https://github.com/JinmiaoChenLab/ezSingleCell2>, I notice the

last update was from 7 months ago, I suppose it was still the old version, thus the code improvement cannot be judged.

Response: Thank you for spotting this and apologies for the delay in updating. We have now updated our Github page.

3. In some step such as “8. Deconvolution”, it shows “Error:....” already before any action. Similar situation happens also in other modules.

Response: Thank you for pointing out this error. We have checked through each module and fixed such errors.

4. Single-cell ATAC-seq pipeline step “10. Gene Set Enrichment Analysis” takes either very long or buggy, please make sure it is still functioning.

Response: Thank you for highlighting this issue. We have fixed the codes and now Gene Set Enrichment Analysis is functioning and runs in a reasonable amount of time. Additionally, we have added one more method – rGREAT (Gu Z and Hübschmann D, 2022) to perform gene set enrichment analysis for scATAC-seq data.

GREAT (Genomic Regions Enrichment of Annotations Tool) uses peaks or region sets as input and associates them with biological functions. GREAT has integrated Gene Ontology (GO) gene sets for more than 600 organisms and MSigDB gene sets (Liberzon *et al.*, 2011) for human. Using such approaches that use peaks (or region sets) as input will be more useful for Gene Set Enrichment Analysis of scATAC-seq data as compared to ‘fgsea’ that uses gene activity as input.

Reviewer #3 (Remarks to the Author):

All my concerns have been addressed.

References:

1. Zuguang Gu, Daniel Hübschmann, rGREAT: an R/bioconductor package for functional enrichment on genomic regions, *Bioinformatics*, Volume 39, Issue 1, January 2023, btac745, <https://doi.org/10.1093/bioinformatics/btac745>
2. Liberzon A. et al. (2011) Molecular signatures database (MSigDB) 3.0. *Bioinformatics*, 27, 1739–1740.

Reviewer #1 (Remarks to the Author):

Most of my concerns have been addressed. Please check some small errors. For example, a long cell type's name cannot be displayed in full on the UMAP, so please consider showing it on two or more rows.